✻ PLOS | ONE

# TMEM98 is a negative regulator of FRAT mediated Wnt/ß-catenin signalling

**Tanne van der Wal[1,2], Jan-Paul Lambooij[3], Renée van Amerongen**[1,2]*

**1** Section of Molecular Cytology, Swammerdam Institute for Life Sciences, University of Amsterdam, Amsterdam, the Netherlands, **2** Van Leeuwenhoek Centre for Advanced Microscopy, University of Amsterdam, Amsterdam, the Netherlands, **3** Division of Molecular Genetics, Netherlands Cancer Institute, Amsterdam, the Netherlands

* r.vanamerongen@uva.nl

## Abstract

Wnt/ß-catenin signalling is crucial for maintaining the balance between cell proliferation and differentiation, both during tissue morphogenesis and in tissue maintenance throughout postnatal life. Whereas the signalling activities of the core Wnt/ß-catenin pathway components are understood in great detail, far less is known about the precise role and regulation of the many different modulators of Wnt/ß-catenin signalling that have been identified to date. Here we describe TMEM98, a putative transmembrane protein of unknown function, as an interaction partner and regulator of the GSK3-binding protein FRAT2. We show that TMEM98 reduces FRAT2 protein levels and, accordingly, inhibits the FRAT2-mediated induction of ß-catenin/TCF signalling. We also characterize the intracellular trafficking of TMEM98 in more detail and show that it is recycled between the plasma membrane and the Golgi. Together, our findings not only reveal a new layer of regulation for Wnt/ß-catenin signalling, but also a new biological activity for TMEM98.

**Data Availability Statement:** All original Yeast-two-hybrid files are available via the Open Science Framework (https://osf.io/ef74w/). All original Western blot files are available via the Open Science Framework (https://osf.io/ef74w/). All

## Introduction

Wnt/ß-catenin signalling is crucial for embryonic development and tissue homeostasis in all multicellular animals. In mammals, it is first required for induction of the primitive streak at the onset of gastrulation [1]. It continues to help control cell proliferation and differentiation at different anatomical sites during all subsequent steps of tissue morphogenesis and throughout postnatal life. At the molecular level, Wnt/ß-catenin signalling promotes the formation of CTNNB1/TCF complexes, in which ß-catenin (CTNNB1) functions as a co-activator for transcription factors of the TCF/LEF family to modulate gene expression in a tissue-specific fashion.

The dynamic expression of 19 different WNT proteins results in a complex signalling landscape [2]. Secreted WNT proteins can interact with a variety of receptors and co-receptors at the cell surface, including FZD, LRP, RYK and ROR [3–6]. Different combinations result in alternative, context-dependent biochemical responses, with Wnt/ß-catenin signalling being just one possible outcome. How these specific cellular responses are induced, both upstream at

other relevant data are within the manuscript and its Supporting Information files.

**Funding:** This work was funded by a MacGillavry fellowship from the University of Amsterdam (to RvA). The funder had no role in study design, data collection and analysis, decision to publish, or preparation of the manuscript.

**Competing interests:** The authors have declared that no competing interests exist.

the level of ligand and receptor binding and downstream at the level of intracellular signal transduction, remains an area of active investigation.

Under physiological conditions, activity of the Wnt/ß-catenin pathway is tightly controlled. A so-called 'destruction complex', containing AXIN1, APC, CSNK1 and GSK3, continuously binds and phosphorylates free CTNNB1. Because phosphorylated CTNNB1 is rapidly degraded by the proteasome, this ensures that low levels of cytoplasmic and nuclear CTNNB1 are maintained in the absence of a WNT signal. When WNT proteins engage FZD/LRP at the cell surface, the destruction complex is inactivated, resulting in an increase in the nucleocyto-plasmic levels of CTNNB1 and the concomitant induction of CTNNB1/TCF transcriptional activity [7–10].

FRAT/GBP proteins are potent activators of CTNNB1/TCF signalling independent from WNT/FZD activity due to their capacity to bind GSK3 [11–13]. FRAT1 and AXIN1 compete for the same binding site on GSK3B [14]. Since the interaction between GSK3 and AXIN1 has been estimated to enhance the phosphorylation of CTNNB1 more than 20,000-fold [15], sequestration of GSK3 by FRAT thus increases CTNNB1/TCF signalling.

First identified as an oncogene in murine T-cell lymphoma progression [16], FRAT1 over-expression indeed correlates with the accumulation of CTNNB1 in a variety of human cancers [17–21]. The *Xenopus* FRAT homologue, GBP, is critically required for dorsoventral axis for-mation as part of the maternal Wnt pathway [13]. However, FRAT function is dispensable for Wnt/ß-catenin signalling in mice [22], indicating that FRAT is a modulator, rather than a core component of the Wnt/ß-catenin pathway in mammals. Moreover, the oncogenic activities of FRAT in lymphomagenesis may at least partially be GSK3 independent [23,24]. To date, the precise role and regulation of FRAT1, and its close homologue FRAT2, remain to be resolved.

Here we identify TMEM98 as a novel FRAT2-binding protein. We show that TMEM98 inhibits FRAT-induced CTNNB1/TCF signalling by reducing FRAT protein levels. We also demonstrate that TMEM98 traffics between multiple endosomal and membrane compart-ments. Together, these findings add a new layer of regulation for Wnt/ß-catenin signalling and provide a potential molecular mechanism for the activities of TMEM98, mutations in which have been linked to autosomal dominant nanophthalmos [25,26].

## Results

### TMEM98 is a novel FRAT2-binding protein

To shed more light on FRAT protein function, we set out to identify new FRAT interactors. Focusing our efforts on FRAT2, we performed a yeast-two-hybrid assay using both full-length FRAT2 and an N-terminal deletion mutant containing the GSK3-binding site (FRAT2ΔN) as a bait. While we did not pick up GSK3 or any other known WNT pathway components in this screen, we did identify a number of putative novel FRAT2 binding proteins (Tables 1 and 2). One candidate, an unknown protein encoded by both the *DKFZp564K1964* and *ETVV536* transcripts, was picked up with high confidence in both the FRAT2 full-length and the FRAT2ΔN screen. We therefore decided to characterize this interaction in more detail.

Since performing these initial studies, the 226 amino-acid FRAT2-binding protein encoded by the *DFKZp564K1964* and *ETVV536* transcripts has officially become annotated as TMEM98, a putative transmembrane protein of unknown function. Similar to FRAT, TMEM98 is highly conserved among vertebrate species, but not present in invertebrates (S1 Fig). The human and mouse homologues are more than 98% identical at the amino acid level, while the human and chick homologues are most divergent, with 73% of amino acid identity (S1 Fig).

**Table 1. Novel FRAT2-binding proteins identified in a yeast-two-hybrid screen with full-length FRAT2 as bait.**

| Protein | Clone Name | # clones | PBS score | Confidence |
|---|---|---|---|---|
| TMEM98 | hDKFZP564K1964 [gi|7661615|ref|NM_015544.1| Homo sapiens DKFZP564K1964 protein (DKFZP564K1964), mRNA] | 5 | A | Very high |
| CEP170 | hkab; hKIAA0470 [gi|7662141|ref|NM_014812.1| Homo sapiens KIAA0470 gene product (KIAA0470), mRNA] | 3 | E | Low |

Putative interactors are listed in order of decreasing confidence (reflected in the Predicted Biological Score (PBS)).

Based on the sequence of the *TMEM98* clones that were isolated in the original yeast-two-hybrid screen, the FRAT2 binding domain of TMEM98 is located in the C-terminal half of the protein, with the longest clone spanning amino acids 66–226 and the shortest clone spanning amino acids 109–216 (S2 Fig). The fact that TMEM98 was identified in both the full-length FRAT2 and the FRAT2ΔN screen suggests that the TMEM98 binding domain of FRAT2 also resides in the C-terminus. By co-expressing myc-tagged FRAT2 and FLAG-tagged TMEM98 plasmid constructs, we were able to confirm binding of FRAT2 and full length TMEM98, but not an N-terminal deletion mutant lacking amino acids 1–34 (TMEM98ΔN), by co-immunoprecipitation from HEK293 cell lysates (Fig 1A and 1B). Western blot analysis revealed TMEM98ΔN to be unstable. Unlike full-length TMEM98, the deletion mutant can only be detected in the presence of the proteasome inhibitor MG132 (Fig 1C). Of note, co-expression of FRAT2 also stabilizes TMEM98ΔN (Fig 1B), suggesting that the two do interact, at least transiently, as a result of which at least some of the TMEM98ΔN escapes degradation.

**Table 2. Novel FRAT2-binding proteins identified in a yeast-two-hybrid screen with FRAT2ΔN as bait.**

| Protein | Clone Name | # clones | PBS score | Confidence |
|---|---|---|---|---|
| CEP170 | hKAB; hKIAA0470; [prey427374—Human KAB] | 15 | A | Very High |
| MRFAP1L1 | hPP784; [gi|44921607|ref|NM_203462.1| Homo sapiens PP784 protein (PP784), transcript variant 2, mRNA] | 11 | A | Very High |
| FTSJ3 | hFTSJ3; [prey427205—Human FTSJ3] | 3 | B | High |
| RALGAPA1 | hGARNL1; [gi|51230411|ref|NM_194301.2| Homo sapiens GTPase activating Rap/RanGAP domain-like 1 (GARNL1), transcript variant 2, mRNA] | 8 | B | High |
| TMEM98 | hETVV536; [gi|37182267|gb|AY358573.1| Homo sapiens clone DNA56050 ETVV536 (UNQ536) mRNA, complete cds] | 5 | B | High |
| RB1CC1 | hRB1CC1; [gi|41350194|ref|NM_014781.3| Homo sapiens RB1-inducible coiled-coil 1 (RB1CC1), mRNA] | 2 | D | Moderate |
| SACS | hSACS; [gi|38230497|ref|NM_014363.3| Homo sapiens spastic ataxia of Charlevoix-Saguenay (sacsin) (SACS), mRNA] | 4 | D | Moderate |
| SEC24C | hSEC24C; [gi|38373668|ref|NM_004922.2| Homo sapiens SEC24 related gene family, member C (S. cerevisiae) (SEC24C), transcript variant 1, mRNA] | 2 | D | Moderate |
| CTNNA2 | hCTNNA2; [prey427324—Human CTNNA2] | 2 | D | Moderate |
| DNM1 | hDNM1; hdynamin; [gi|181848|gb|L07807.1|HUMDYNA Human dynamin mRNA, alternative exons and complete cds] | 4 | D | Moderate |
| FBXO30 | hFBXO30; [gi|54112383|ref|NM_032145.4| Homo sapiens F-box protein 30 (FBXO30), mRNA] | 3 | D | Moderate |
| ATRX | hATRX; [gi|20336208|ref|NM_000489.2| Homo sapiens alpha thalassemia/mental retardation syndrome X-linked (RAD54 homolog, S. cerevisiae) (ATRX), transcript variant 1, mRNA] | 6 | E | Low |
| UBAC1 | hGDBR1; hGBDR1; hUBADC1; [gi|7705380|ref|NM_016172.1| Homo sapiens ubiquitin associated domain containing 1 (UBADC1), mRNA] | 2 | E | Low |

Putative interactors are listed in order of decreasing confidence (reflected in the PBS score).

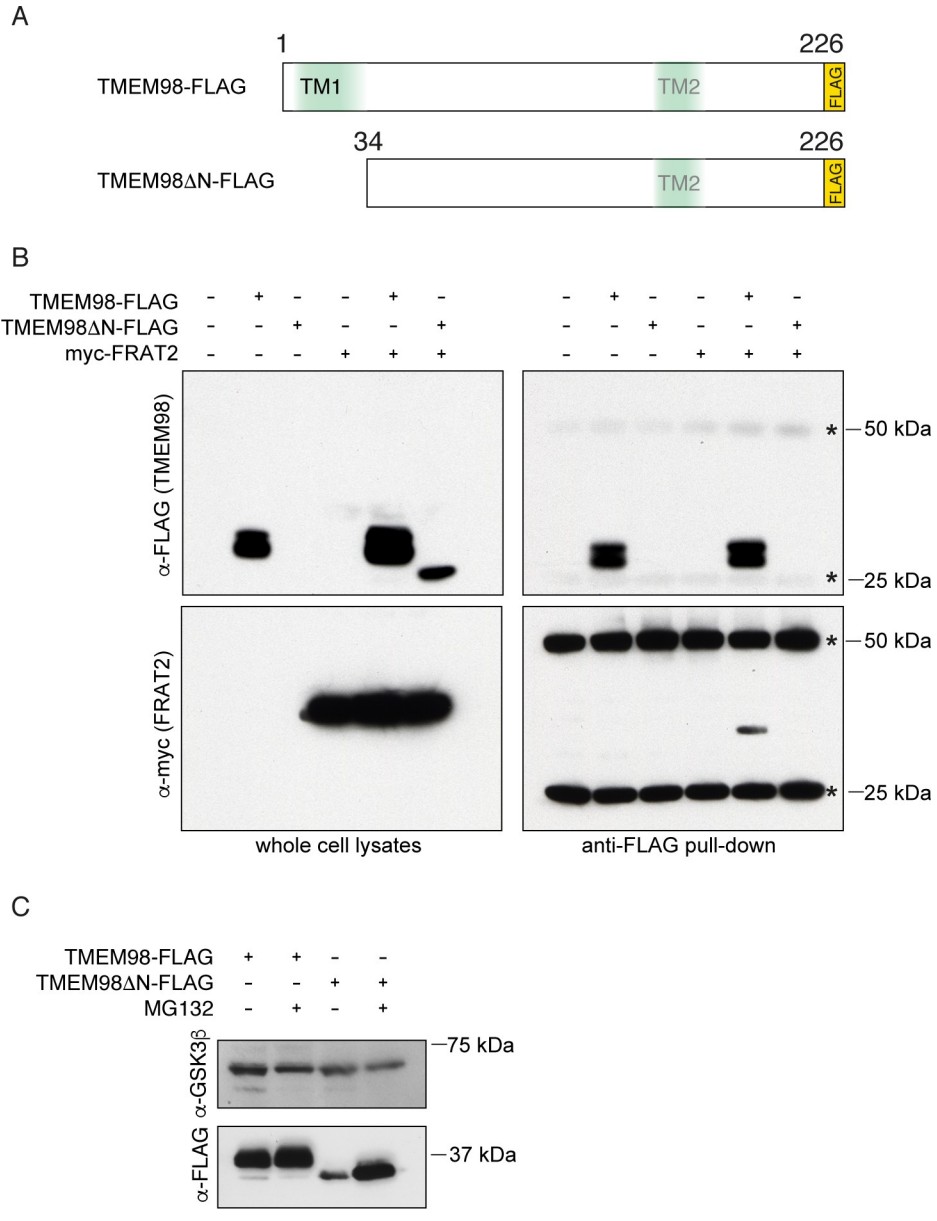

**Fig 1. TMEM98 binds FRAT2.** (A) Schematic showing FLAG-tagged expression constructs of full-length TMEM98 (amino acids 1–226) and an N-terminal deletion mutant (amino acids 34–226, TMEM98ΔN). Topology prediction programs indicate a potential signal sequence or N-terminal transmembrane region (TM1) and a putative second transmembrane region (TM2) around position 161–172. (B) Western blot showing co-immunoprecipitation of myc-FRAT2 with full-length TMEM98-FLAG in lysates from transiently transfected HEK239T cells. Asterisks indicate cross reactivity of the secondary antibody with the heavy and light chain of the anti-FLAG antibody used to pull down TMEM98-FLAG. The deletion mutant TMEM98ΔN-FLAG is not pulled down under the conditions used. However, it can be detected in protein lysates when myc-FRAT2 is co-transfected. Size markers are indicated. (C) Western blot showing a stabilizing effect of the proteasome inhibitor MG132 on TMEM98ΔN-FLAG protein levels following transient transfection of the indicated constructs in HEK293T cells. Endogenous GSK3ß was used as a loading control. Size markers are indicated.

## TMEM98 membrane localization and topology

Because TMEM98 did not contain any known motifs or functional sites, we performed hydrophobicity analyses and subcellular localization predictions. All five secondary structure

**Table 3. Secondary structure prediction of TMEM98.**

| Algorithm | TM regions | Topology | Other |
|---|---|---|---|
| *DAS-TMfilter* | 6–25 (TM1) | n.p. | Potential signal peptide in TM1 |
| | 161–172 (TM2) | | |
| *HMMTOP* | 9–32 (TM1) | N-terminus out; Single pass type I protein | |
| *PHOBIUS* | 6–27 (TM1) | N-terminus out; Single pass type I protein | Non-significant signal peptide in TM1; Non-significant TM2 around 156–173 |
| *TMHMM* | 4–26 (TM1) | N-terminus out; Single pass type I protein | Potential N-terminal signal peptide |
| *TMpred* | 4–26 (TM1) | N-terminus out; Single pass type I protein | Non-significant TM2 around 157–176 |

Predictions are based on the human TMEM98 primary amino acid sequence. Putative transmembrane regions are numbered according to their amino acid position. TM = transmembrane, n.p. = not predicted.

prediction algorithms used (HMMTOP; Phobius; TMHMM; TMpred; DAS-TMfilter) unanimously predict TMEM98 to have an N-terminal transmembrane domain spanning residues 6–25 (Table 3, S1 Fig). A TMEM98-GFP fusion protein indeed localizes to the plasma membrane (Fig 2A), in addition to showing more prominent localization to the Golgi and intracellular vesicles (Fig 2A and 2B). Forward trafficking was inhibited by treatment of the cells with Brefeldin A, a reversible inhibitor of protein transport from the ER to the Golgi (Fig 2C), suggesting that TMEM98 is transported to the plasma membrane via the classical, secretory pathway.

To test the requirement of the TMEM98 N-terminus for targeting, we compared the localization of TMEM98 and TMEM98ΔN. Whereas full-length TMEM98 localizes to the Golgi and the plasma membrane, TMEM98ΔN shows diffuse expression throughout the cytoplasm at much lower levels (Fig 2D), in agreement with its rapid turnover by the proteasome (Fig 1C). Further support for a role of the TMEM98 N-terminus in subcellular targeting comes from the fact that an N-terminally tagged version of TMEM98 (GFP-ORF) becomes trapped in the ER, in contrast to a C-terminal fusion protein (ORF-GFP) that properly traffics to the Golgi and the cell membrane (S3 Fig, [27–29]). Finally, the first 34 amino acids by themselves (TM1) are sufficient to target a fluorescent protein to the ER (Fig 2E). Together, these results demonstrate that the N-terminus of TMEM98 is both necessary and sufficient for targeting to the secretory pathway.

Although these results establish that TMEM98 contains an N-terminal transmembrane domain, prediction algorithms and the published literature do not agree on its exact topology or subcellular localization. Two out of five secondary structure prediction algorithms propose the presence of a putative N-terminal signal peptide (TMHMM; DAS-TMfilter, Table 3), but only one signal peptide prediction algorithm also estimates this sequence to be cleaved (PrediSi, Table 4). Four out of five algorithms (HMMTOP; Phobius; TMHMM; TMpred) predict TMEM98 to be a single pass type I protein (Table 3). In contrast, an experimental study reported TMEM98 to be a single pass type II protein, which was also found to be secreted in its full-length form [30]. To resolve this apparent discrepancy, we first determined whether the C-terminus of TMEM98 is located extracellularly or intracellularly (Fig 2F). To this end, we generated a stable cell line expressing a doxycycline inducible TMEM98-FLAG protein (Fig 2G) and stained these cells with an anti-FLAG antibody under both unpermeabilized and permeabilized conditions. Subsequent FACS analysis revealed a clear signal for doxycycline-induced cells in both conditions (Fig 2H, S4 Fig), thus confirming that the TMEM98 C-terminus is, at least partially, located on the extracellular side of the plasma membrane.

## TMEM98 is a negative regulator of Wnt/ß-catenin signalling

Having confirmed that FRAT2 and TMEM98 physically interact (Fig 1), we next investigated whether TMEM98 altered FRAT2 activity. To this end, we measured FRAT2 induced

CTNNB1/TCF signalling in HEK293T cells using the well-known TOPFLASH luciferase reporter assay. Co-transfection experiments showed that TMEM98 reduced the level of CTNNB1/TCF signalling induced by FRAT2 (Fig 3A).

We next asked whether the observed effect was specific for FRAT2. To this end, we measured the effect of TMEM98 on the induction of CTNNB1/TCF signalling by both myc-FRAT1 and myc-FRAT2 (Fig 3B). Full-length TMEM98 almost reverted FRAT2-induced TOPFLASH reporter activity back to baseline (from 4.5 ± 1.8 to 1.1 ± 0.48 fold induction; mean ± standard deviation). When equal amounts of *FRAT1* plasmid DNA were transfected, full-length TMEM98 reduced FRAT1-mediated TOPFLASH reporter activity to a similar extent (from 9.9 ± 6.1 to 3.3 ± 2.3 fold induction), equating to a 66% reduction for FRAT1 and a 76% reduction for FRAT2 on average. Indeed, myc-FRAT1 can also be co-immunoprecipitated with full length TMEM98-FLAG upon overexpression in HEK293T cells (S5 Fig). Thus, TMEM98 is capable of interacting with both FRAT homologues and blocks FRAT activity.

## TMEM98 reduces FRAT2 protein levels

To determine how TMEM98 inhibits the signalling activities of FRAT2, we tested whether TMEM98 affected FRAT2 protein levels. To this end, we co-transfected a constant amount of *myc-Frat2* plasmid DNA with increasing amounts of *Tmem98* in HEK293T cells. Quantitative Western blot analysis revealed a variable, but consistent reduction in FRAT2 protein levels in the presence of full length TMEM98 (Fig 4A and 4B and S7A–S7C Fig).

To further substantiate that TMEM98 controls FRAT protein levels, we knocked down *Tmem98* expression, reasoning that this should at least partially restore FRAT protein levels. Of the three RNAi constructs designed, two efficiently reduced both TMEM98 and TMEM98ΔN protein levels (S6 Fig). Western blot analysis showed that both FRAT1 and FRAT2 protein levels indeed increased when either *Tmem98-FLAG* or *Tmem98ΔN-FLAG* expression was knocked down (Fig 4C and 4D). The RNAi constructs were designed to also recognize endogenous *TMEM98* in HEK293T cells. Importantly, knocking down endogenous TMEM98 expression also resulted in an increase in FRAT1 and FRAT2 protein levels (Fig 4C and 4D). This translated to an increase in TOPFLASH reporter activity (Fig 4E and 4F), with our most efficient knockdown construct (*TMEM98 RNAi #3*) causing an average 1.7-fold and 1.6-fold increase in TOPFLASH induction by myc-FRAT1 (from 15.3 ± 8.1 to 26.7 ± 6.0) and myc-FRAT2 (from 5.4 ± 3.3 to 8.5 ± 3.1), respectively. Together, these results confirm that TMEM98 inhibits FRAT function by reducing FRAT protein levels.

## TMEM98 undergoes retrograde trafficking

A previous study showed TMEM98 to be secreted in its full-length form, presumably via exosomes [30]. Whereas exosomes originate from multivesicular bodies (MVBs), our results suggest that TMEM98 reaches the plasma membrane via the classical secretory pathway (i.e. via the ER/Golgi, Fig 2).

However, when we analysed publicly available protein-protein interaction data using the BioGRID tool [31,32], we noticed an interaction between TMEM98 and multiple proteins that are associated with late (STX7, STX8, VT1B, VAMP8) and/or recycling endosomes (STX6, STX12, VAMP3). In addition, TMEM98 was also picked up in a yeast-two-hybrid screen as a binding partner of RABEPK, a Rab9 effector that is required for transport from endosomes to the trans-Golgi network [33,34]. Although none of these interactions have been verified in living cells thus far, these findings suggest that TMEM98 may undergo retrograde trafficking (S8 Fig). Using confocal microscopy, we found that the TMEM98-GFP signal partially overlapped

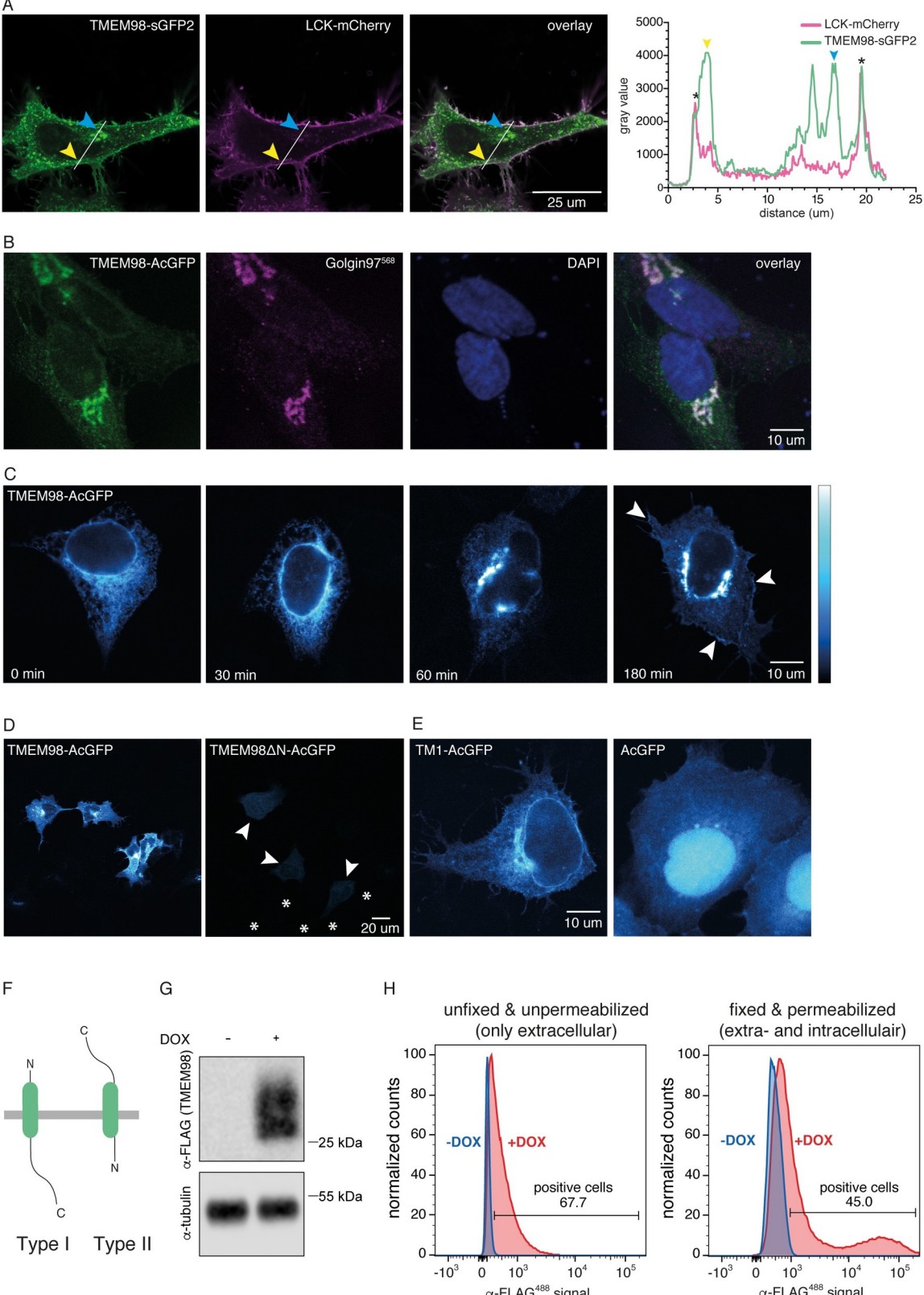

**Fig 2. TMEM98 is a transmembrane protein with an N-terminal transmembrane domain and an extracellular C-terminus.** (A) Live-cell confocal microscopy image of HeLa cells co-transfected with TMEM98-sGFP2 (left) and LCK-mCherry (which localizes to the

plasma membrane, middle), showing co-localization of the two proteins in the membrane (right, overlay). The yellow and blue arrowheads indicate prominent localization of TMEM98, but not LCK, to large vesicular structures (yellow) and the Golgi (blue). Graph depicts the intensity plot profile of the fluorescent signal for TMEM98-sGFP2 and LCK-mCherry along the line drawn in the panels on the left. Yellow and blue arrowheads serve as points of reference. Asterisks indicate the plasma membrane. (B) Confocal microscopy image of methanol-fixed HeLa cells transiently transfected with TMEM98-AcGFP (left). Staining with anti-Golgin97 antibody (middle) shows that TMEM98-AcGFP localizes to the Golgi apparatus (right, overlay). Nuclei are stained with DAPI and are shown in blue. Note that membrane localization is less prominent in this case because the Golgi is located in a different, more apical focal plane. (C) Pseudo-coloured confocal microscopy images of fixed HEK293A cells transiently transfected with TMEM98-AcGFP, showing that TMEM98 travels to the plasma membrane via the secretory route. Cells were treated with Brefeldin A for 4 hours to block forward trafficking through the Golgi, after which they were released. Cells were fixed at 0, 30, 60 and 180 minutes following release. Representative examples are shown. After 60 minutes, TMEM98-AcGFP reappears in the Golgi and after 180 minutes some signal can again be detected at the plasma membrane (arrowheads). (D) Pseudo-coloured confocal microscopy images of fixed HEK293A cells transiently transfected with TMEM98-AcGFP (left) or TMEM98ΔN-AcGFP, a deletion mutant lacking the first 34 amino acids (right). TMEM98ΔN-AcGFP localizes throughout the cell at much lower levels than the full length fusion protein (individual cells are indicated by arrowheads, some non-transfected surrounding cells are indicated by an asterisk positioned in the nucleus). (E) Pseudo-coloured confocal microscopy images of fixed 293A cells transiently transfected with a fusion construct in which AcGFP was tagged with the first 34 amino acids of TMEM98 (TM1-AcGFP, left), showing that TM1 is sufficient for targeting to the ER. In contrast, an untagged, free-floating fluorescent protein diffuses throughout the cytoplasm and nucleus (AcGFP, right). (F) Schematic depicting the orientation of a single-pass type I (left) or type II (right) transmembrane protein. (G) Western blot showing doxycycline dependent induction of TMEM98-FLAG in HEK293T cells stably transfected with *tetO-TMEM98-FLAG* and *CMV-rtTA3*. Size markers are indicated. (H) FACS analysis of the cells described in (G) in the absence (blue) and presence (red) of doxycycline (DOX) to induce TMEM98-FLAG expression. Following staining with an anti-FLAG antibody, TMEM98-FLAG can be detected in both unfixed, unpermeablized cells (left) and in fixed, permeabilized cells (right). The higher signal obtained in fixed, permeabilized cells likely reflects the large pool of TMEM98-FLAG present in the Golgi and intracellular vesicles, which is not detected by only staining the plasma membrane pool with an extracellularly exposed C-terminus.

with that of the early endosome marker EEA1 (Fig 5A), a RAB5 effector and binding partner of STX6 [35,36]. From this, we conclude that TMEM98 indeed enters the endocytic pathway.

To further follow its intracellular trafficking, we first tested whether TMEM98 reaches the lysosomal compartment. The signal of TMEM98-mTq2, which is still fluorescent at low intracellular pH [37] only occasionally overlapped with that of a lysosomal dye (Fig 5B). In addition, although TMEM98-positive vesicles were frequently found in close proximity to LAMP1-positive lysosomes, we were ultimately unable to confidently determine that the two indeed co-localized (Fig 5C). TMEM98-FLAG protein turnover was unchanged by the lysosomal inhibitor bafilomycin (S9 Fig). Thus, we were unable to find conclusive evidence for either lysosomal targeting or degradation of TMEM98.

**Table 4. Subcellular localization predictions of TMEM98.**

| Algorithm | Cleavage position | Subcellular localization | Score/Confidence |
|---|---|---|---|
| *PrediSi* | 29 | secreted | 0.6445 |
| *WOLF PSORT* | n.p. | Extracellular | 21 |
| | | ER | 5 |
| | | Plasma membrane | 3 |
| *SherLoc2* | n.p. | ER | 0.57 |
| | | Golgi | 0.25 |
| | | Extracellular | 0.07 |
| | | Plasma membrane | 0.05 |
| | | Lysosomal | 0.05 |
| | | Cytoplasmic | 0.01 |
| | | Mitochondrial | 0.0 |
| | | Nuclear | 0.0 |
| | | Peroxisomal | 0.0 |

Predictions are based on the human TMEM98 primary amino acid sequence. Putative cleavage sites are numbered according to their amino acid position. n.p. = not predicted.

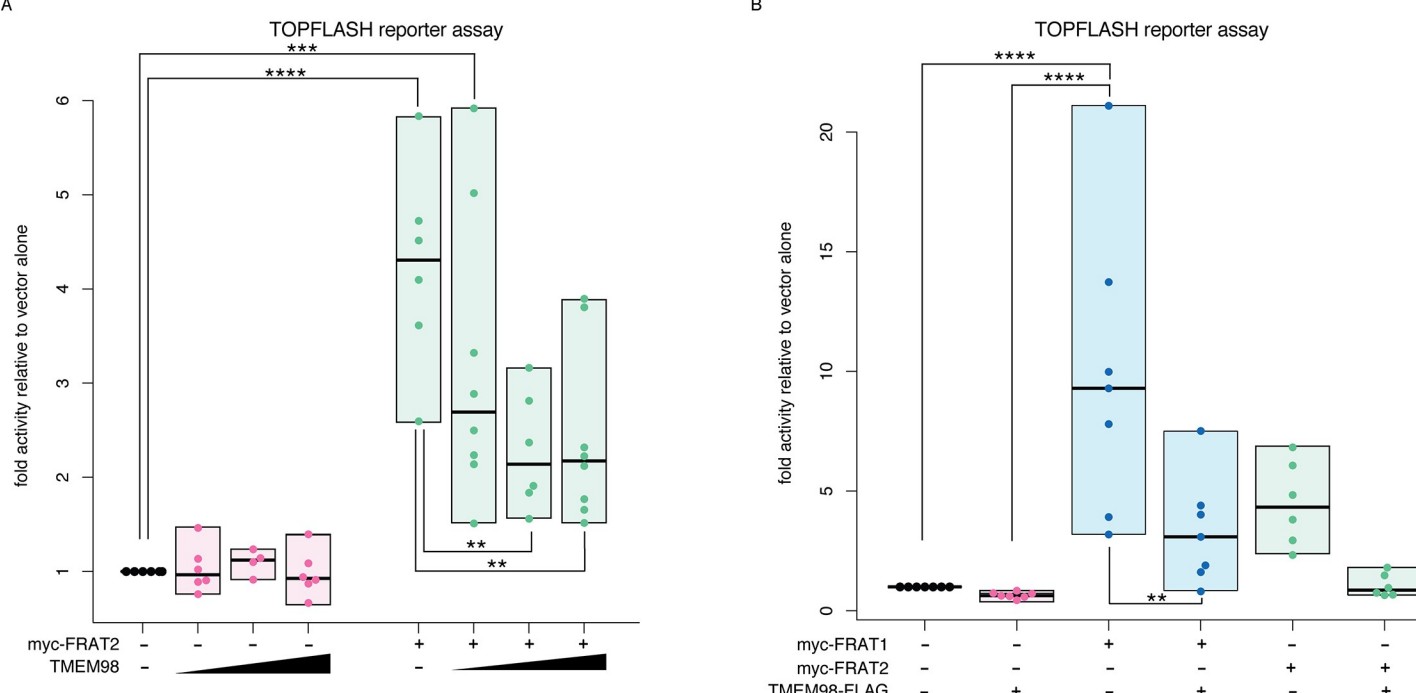

**Fig 3. TMEM98 inhibits FRAT-induced CTNNB1/TCF signalling.** (A) Dual luciferase reporter assay in transiently transfected HEK293T cells to quantify the levels of CTNNB1/TCF signalling. Transfecting increasing amounts of Tmem98-FLAG cDNA with a constant amount of myc-Frat2 cDNA inhibits FRAT2 induced TCF/LEF (TOPFLASH) reporter activity. Graph depicts data from n = 4–8 independent experiments for each condition, with each data point representing the average of three technical replicates. Boxes depict the spread of the data. Horizontal bars depict the median value. TOPFLASH luciferase values are normalized to CMV-Renilla, which was included as a transfection control. For each individual experiment the baseline TOPFLASH luciferase reporter activity in HEK293T cells transfected with empty vector (instead of myc-Frat2) was set to 1. Statistically significant differences (one-way ANOVA, Tukey's multiple comparisons test) are indicated with asterisks: ****: $P \leq 0.0001$, ***: $P \leq 0.001$, **: $P \leq 0.01$, *: $P \leq 0.05$. (B) Same as in (A), but this time showing the effect of full-length TMEM98-Flag on myc-FRAT1 as well as myc-FRAT2. Note that FRAT1 is a more potent activator of CTNNB1/TCF signalling than FRAT2. Graph depicts data from n = 6–7 independent experiments, with each data point representing the average of three technical replicates. Boxes depict the spread of the data. Horizontal bars depict the median value. Only statistically significant differences (one-way ANOVA, Tukey's multiple comparisons test) are indicated with asterisks: ****: $P \leq 0.0001$, ***: $P \leq 0.001$, **: $P \leq 0.01$, *: $P \leq 0.05$.

While performing these experiments, we noticed that many of the intracellular TMEM98-positive structures had a more tubular appearance (Fig 5D and other examples), which is in agreement with that of endosomal recycling domains [38–40]. Indeed, we detected TMEM98-mTq2 in multiple endosomal compartments, as evidenced by its co-localization with RAB9A (involved in retrograde transport to the trans-Golgi network, Fig 5E) and RAB7 (present on maturing and late endosomes, Fig 5F). In addition, we found the TMEM98-GFP signal to also show partial overlap with that of the mannose-6-phosphate receptor (MPR), which is recycled to the trans-Golgi network from late endosomes in a RAB9-dependent manner (Fig 5G) [41]. Together, our data suggest that TMEM98 traffics between multiple endosomal compartments, is recycled between the Golgi and the plasma membrane, but largely escapes lysosomal degradation.

## Discussion

Wnt signal transduction is tightly controlled. Multiple agonists and antagonists modify activity of the Wnt/ß-catenin pathway at the ligand and receptor level, with DKK and RSPO serving as prominent examples [42–48]. Wnt-signalling strength can also be modulated intracellularly. For instance, expression of the negative feedback regulator *Axin2* is induced in virtually all cells with active Wnt/ß-catenin signalling [49,50]. This ensures re-association of the

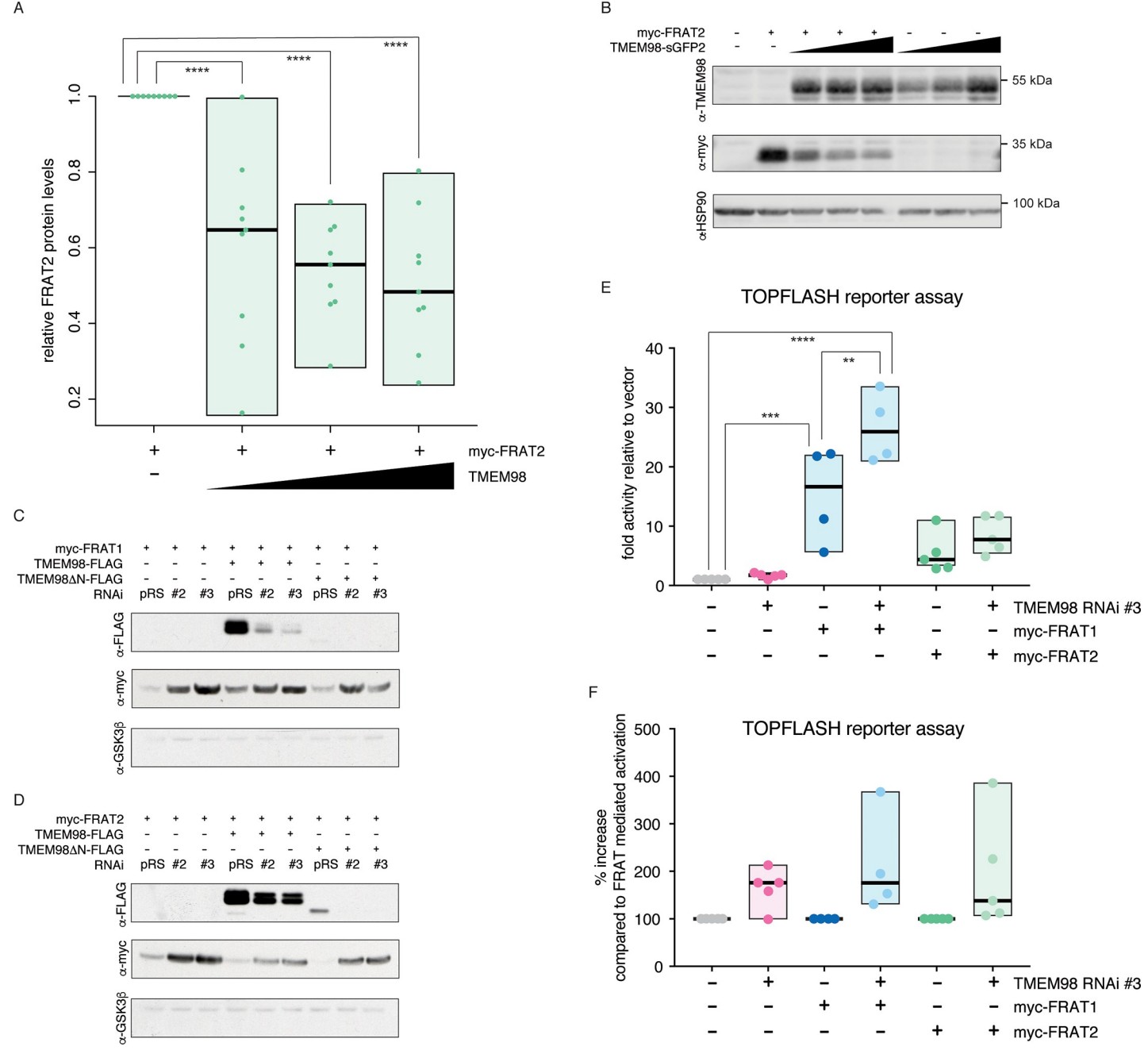

**Fig 4. TMEM98 negatively regulates FRAT protein levels.** (A) Quantification of myc-FRAT2 protein levels upon co-transfecting increasing concentrations of TMEM98-sGFP2 or TMEM98-mTq2 fusion constructs, confirming that increasing concentrations of TMEM98 result in a reduction in myc-FRAT2 protein levels. Graph depicts data from n = 9 independent experiments. Boxes depict the spread of the data. Horizontal bars depict the median value. For each experiment, myc-FRAT2 protein levels in the absence of TMEM98 were set to 1. (B) Representative example of a Western blot from the experiments quantified in (A). HSP90 serves as a loading control. (C-D) Western blot of lysates from transiently transfected HEK293T cells, showing that knocking down either endogenous (first three lanes) or transiently transfected TMEM98-FLAG (next three lanes) and TMEM98ΔN-FLAG (last three lanes) results in an increase in both myc-FRAT1 (C) and myc-FRAT2 (D) protein levels. pRS = pRetrosuper, the empty vector control for RNAi constructs #2 and #3. Endogenous GSK3ß serves as a loading control. (E) Dual luciferase reporter assay in transiently transfected HEK293T cells, quantifying the effects of knocking down endogenous Tmem98 using the most efficient RNAi (#3) on FRAT induced TOPFLASH reporter activity. Graph depicts data from n = 4 (FRAT1) or n = 5 (FRAT2) independent experiments, with each data point representing the average of three technical replicates. Boxes depict the spread of the data. Horizontal bars depict the median value. TOPFLASH luciferase values are normalized to CMV-Renilla, which was included as a transfection control. For each individual experiment the baseline TOPFLASH luciferase reporter activity in HEK293T cells transfected with empty vector (instead of myc-Frat1 or myc-Frat2) was set to 1. (F) Same as in (E), but this time the TOPFLASH luciferase reporter activation in the absence of a Tmem98 knockdown was set to 100%. Only statistically significant differences (one-way ANOVA, Tukey's multiple comparisons test) are indicated with asterisks: ****: P ≤ 0.0001, ***: P ≤ 0.001, **: P ≤ 0.01.

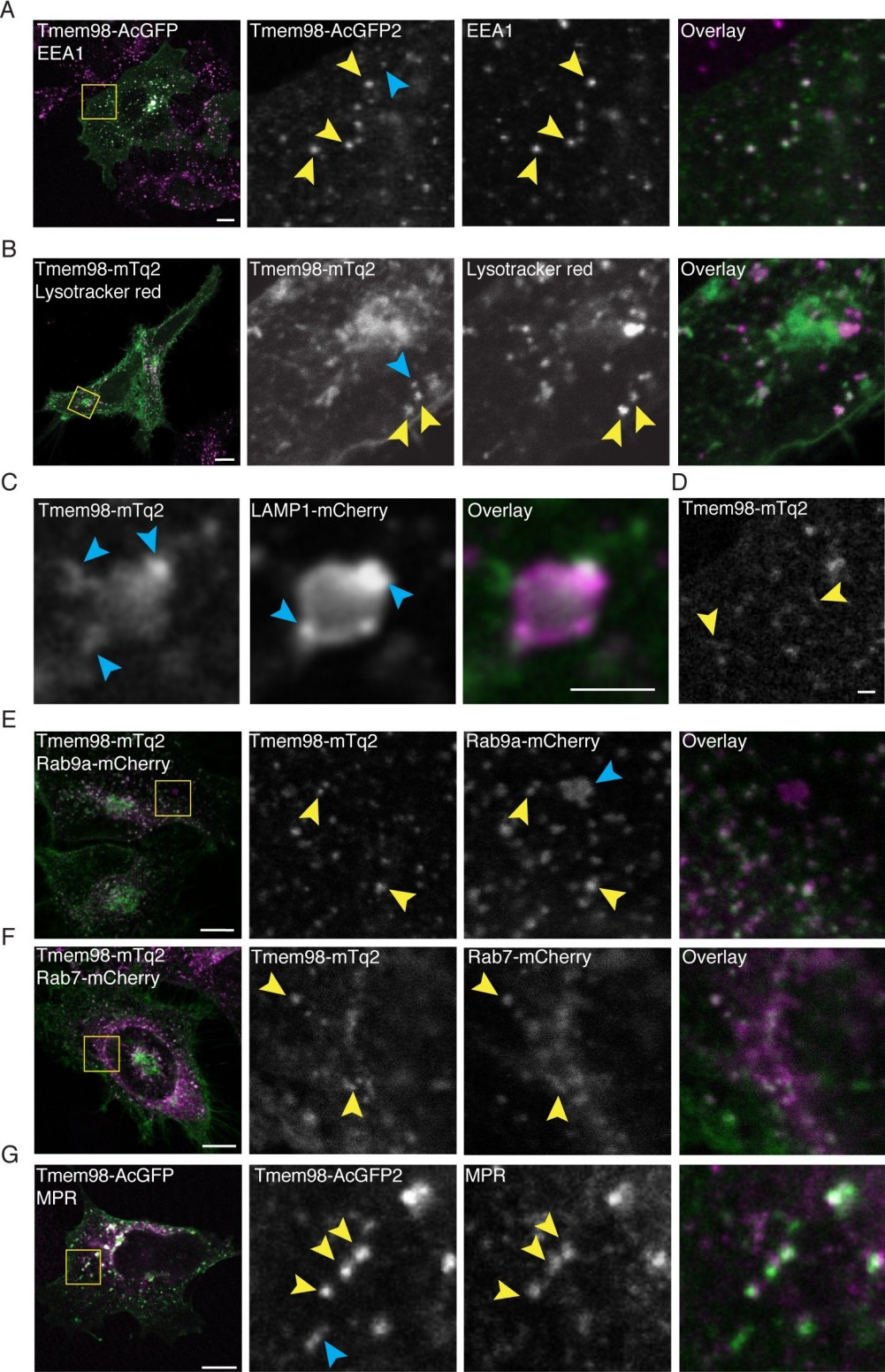

**Fig 5. TMEM98 is recycled between the Golgi and the plasma membrane.** (A) Confocal microscopy images of fixed 293A cells, showing co-localization of TMEM98 and early endosomes. Green: Transiently transfected TMEM98-AcGFP (direct detection of GFP signal). Magenta: Immunofluorescent staining of endogenous EEA1. Scale bar is 10 μm. (B) Confocal microscopy images of live HeLa cells, showing partial overlap of TMEM98 and lysosomes. Green: Transiently

transfected TMEM98-mTq2 (direct detection of mTq2 signal). Magenta: Lysotracker red dye (direct fluorescent detection). Scale bar is 10 µm. (C) Confocal microscopy images of live HeLa cells, showing close proximity but incomplete overlap of TMEM98 and lysosomes. Green: Transiently transfected TMEM98-mTq2 (direct detection of mTq2 signal). Magenta: Transiently transfected LAMP1-mCherry (direct detection of mCherry signal). Scale bar is 2 µm. (D) Confocal microscopy image of fixed HeLa cells, transiently transfected with TMEM98-mTq2 and highlighting the tubular appearance of TMEM98-positive vesicular structures (direct detection of mTq2 signal). Scale bar is 1 µm. (E) Confocal microscopy images of live HeLa cells, showing co-localization of TMEM98 and recycling endosomes. Green: Transiently transfected TMEM98-mTq2 (direct detection of mTq2 signal). Magenta: Transiently transfected Rab9a-mCherry (direct detection of mCherry signal). Scale bar is 10 µm. (F) Confocal microscopy images of live HeLa cells, showing co-localization of TMEM98 and late endosomes. Green: Transiently transfected TMEM98-mTq2 (direct detection of mTq2 signal). Magenta: Transiently transfected Rab7-mCherry (direct detection of mCherry signal). Scale bar is 10 µm. (G) Confocal microscopy images of fixed 293A cells, showing co-localization of TMEM98 and late endosomes. Green: Transiently transfected TMEM98-AcGFP (direct detection of GFP signal). Magenta: Immunofluorescent staining of endogenous MPR. Scale bar is 10 µm. (A-G) Yellow arrowheads indicate co-localization, blue arrowheads indicate no co-localization.

destruction complex and dampening of the CTNNB1/TCF signalling response. In contrast, FRAT proteins are positive regulators of the Wnt/ß-catenin pathway. Their GSK3-binding activity allows initiation or amplification of the CTNNB1/TCF response even in the absence of a WNT stimulus. However, little remains known about their physiological function or their precise regulation at the molecular level.

Here we characterize TMEM98 as a novel negative regulator of FRAT and an inhibitor of FRAT-induced CTNNB1/TCF signalling. TMEM98 binds FRAT2, resulting in a reduction in FRAT2 protein levels and a concomitant decrease in FRAT2 signalling activity (Figs 1, 3 and 4). Because TMEM98 protein levels did not increase linearly in the presence of FRAT2, as opposed to when TMEM98 was transfected alone (Fig 4B and S7A–S7D Fig), our findings are consistent with a model in which TMEM98 and FRAT2 form a negative feedback loop (S7E Fig). As such, TMEM98 and FRAT2 may constitute a dynamic regulatory switch with the capacity to fine-tune CTNNB1/TCF signalling activity. TMEM98 also binds and inhibits FRAT1, but whether TMEM98 affects the biological activity of both homologues to the same extent remains to be determined.

Our experiments largely agree with topology predictions and previous findings by confirming that TMEM98 localizes to the plasma membrane via an N-terminal membrane anchor (Fig 2). However, where most algorithms predict TMEM98 to be a single-pass type I transmembrane protein with an intracellular C-terminus, our results confirm that TMEM98 has an extracellular C-terminus, as proposed previously [51]. From our yeast-two-hybrid screen we can conclude that amino acids 109–217 of TMEM98 interact with FRAT2 (S2 Fig). Given that FRAT2, as far as we know, is a soluble cytoplasmic protein, these findings raise an interesting conundrum: if TMEM98 indeed has an extracellular C-terminus, then where does it encounter and interact with FRAT? We present two possible scenarios (Fig 6).

If TMEM98 is a single-pass type II protein with its entire C-terminus located extracellularly, logic dictates that the only intracellular location where interaction between TMEM98 and FRAT could take place would be inside early or late endosomes (Fig 6, Model A), the contents of which are either recycled, targeted for lysosomal degradation or secreted in exosomes. Although neither FRAT1 nor FRAT2 have yet been shown to localize to the endosomal compartment, other WNT/ß-catenin signalling components, including GSK3, have [56–58]. Obviously, targeting of FRAT to either lysosomes or exosomes would result in a reduction of FRAT protein levels. Although we have so far not been able to find experimental support for either of these scenarios, our immunoprecipitation experiments support the fact that only a small proportion of the total FRAT2 protein pool interacts with TMEM98 (Fig 1). Alternatively, the presence of a second transmembrane domain in TMEM98 would create and intracellular loop

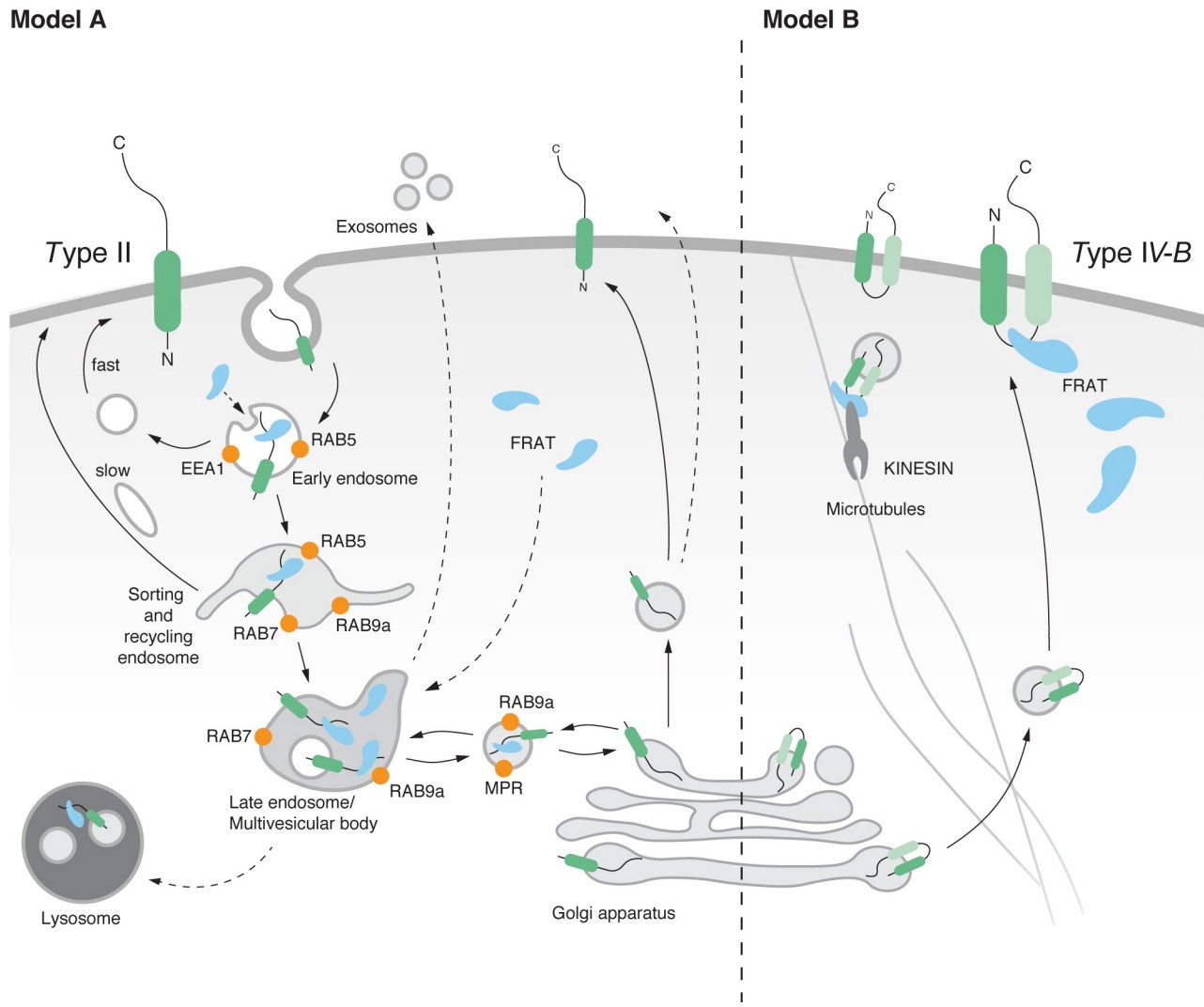

**Fig 6. Model for FRAT and TMEM98 interaction.** Schematic representation of possible TMEM98 topologies and trafficking activities based on in silico prediction algorithms and our experimental data. Given that most algorithms predict TMEM98 to be a single-pass transmembrane protein and our data confirm that the C-terminus of the protein is located on the extracellular surface, model A (left) is the most likely scenario. However, it would require TMEM98 and FRAT to interact in the endosomal compartment, for which no evidence is available to date. The alternative model B (right) assumes the presence of a second transmembrane domain, to allow exposure of a cytosolic FRAT-binding domain. Depiction of the endosomal trafficking and sorting compartments is based on the previously reported association of different Rab proteins with specific endosomal domains [40,52–55].

and potential binding site that would be capable of interacting with the cytoplasmic pool of FRAT (Fig 6, Model B). Of note, the DAS-TMfilter algorithm indicates the presence of a second, more C-terminal, transmembrane domain (TM2, Fig 1A, S1C Fig and Table 3) and, accordingly, predicts TMEM98 to be a dual pass transmembrane protein. Other algorithms (Phobius & TMpred) also detect this domain, but indicate it as non-significant. However, it should be noted that a previous study detected TMEM98 on the extracellular cell surface using an antibody with an epitope immediately downstream of the N-terminal transmembrane helix [30]. This makes the existence of such an intracellular, cytoplasmic loop less likely.

Interestingly, the *Xenopus* FRAT homologue, GBP, has previously been found to bind to Kinesin light chain (KLC) and, as such, to be transported along microtubules during the

process of cortical rotation in early frog embryogenesis [59]. It is tempting to speculate that FRAT and TMEM98 could be similarly transported. Perhaps they are more broadly involved in endocytic trafficking in mammalian cells, which is also known to occur along microtubules in a kinesin-dependent manner [60]. Of course, we cannot exclude other possibilities. Theoretically, TMEM98 can also exist in two different confirmations: a single pass type I and type II protein that could flip its orientation at the plasma membrane, as has been shown to occur upon changes in lipid composition [61]. Which, if any, of these models is correct and how this would ultimately result in a reduction in FRAT protein levels, remains to be tested.

Both FRAT and TMEM98 have been shown to have oncogenic activity [16,23,24,62,63]. While the physiological role of FRAT proteins is still elusive [22], TMEM98 has been genetically linked to autosomal dominant nanophthalmos [25,26], a developmental disorder resulting in small eyes, hyperopia and an increased risk of angle closure glaucoma. A molecular and cellular explanation for the involvement of TMEM98 in eye development and the onset of nanophthalmos is still missing. Of note, TMEM98 was recently reported to bind and prevent the self-cleavage of MYRF, an ER associated membrane-bound transcription factor [64]. MYRF itself has since been linked to nanophthalmos in humans and retinal degeneration in mice [65], similar to TMEM98 [66]. It will be interesting to determine if and how the FRAT-binding and vesicular trafficking activities of TMEM98 contribute to its biological function in this setting.

## Materials and methods

### Yeast two-hybrid screening

Full-length murine *Frat2* (encoding amino acids 1–232) or a deletion mutant starting at an internal SmaI site (*Frat2ΔN*, encoding amino acids 139–232) were cloned in a *LexA* C-terminal fusion vector provided by Hybrigenics. Constructs were confirmed to be in frame by Sanger sequencing.

Yeast two-hybrid (Y2H) screening was performed by Hybrigenics, S.A., Paris, France (http://www.hybrigenics.com). Shortly, the bait construct was transformed in the L40ΔGAL4 yeast strain [67]. A Human fetal brain random-primed cDNA library, transformed into the Y187 yeast strain and containing ten million independent fragments, was used for mating. The screen was first performed on a small scale to adapt the selective pressure to the intrinsic property of the bait. For full-length *Frat2*, no auto-activation of the bait was observed. The *Frat2ΔN* bait was found to auto-activate the Y2H system, and 50mM 3-aminotriazole was found to be the optimal concentration to reduce background colonies. Next, the full-scale screen was performed in conditions ensuring a minimum of 50 million interactions tested, in order to cover five times the primary complexity of the yeast-transformed cDNA library [68]. A total of 95 million (*Frat2* screen) and 87 million (*Frat2ΔN* screen) interactions were actually tested. After selection on medium lacking leucine, tryptophane, and histidine, 8 and 141 positive clones were picked for *Frat2* and *Frat2ΔN*, respectively, and the corresponding prey fragments were amplified by PCR and sequenced at their 5' and 3' junctions.

Sequences were then filtered as described previously [69] and compared to the latest release of the GenBank database using BLASTN [70]. A Predicted Biological Score (PBS) was assigned to assess the reliability of each interaction [67,68]. First, a local score takes into account the redundancy and independency of prey fragments, as well as the distributions of reading frames and stop codons in overlapping fragments. Second, a global score takes into account the interactions found in all the screens performed at Hybrigenics using the same library. In addition, potential false-positives are flagged by a specific "E" PBS score. This is done by discriminating prey proteins containing "highly connected" domains, previously found several times in

screens performed on libraries derived from the same organism. Raw data of the yeast-two-hybrid screen are available via the Open Science Framework at http://dx.doi.org/10.17605/OSF.IO/EF74W.

### DNA constructs

The *pGlomyc-Frat1* and *pGlomyc-Frat2* plasmids were described previously [11,12]. The full-length coding sequence of murine *Tmem98* was amplified from a mix of murine embryonic and murine keratinocyte cDNA using forward 5′-AAAAAGCTTGCCATGGAGACTGTGGTGATCGTC-3′ and reverse 5′-TTTTGAATTCTTAAATGGCCGACTGTTCCTGCAGGAAGC-3′ primers and cloned into pSP72 (Promega) as a HindIII/EcoRI fragment. A *FLAG* tag was inserted at the C-terminus via a PstI/EcoRI restriction digest. The *Tmem98-FLAG* cassette was cloned into pCDNA3.1 as a HindIII/HindIII fragment. The deletion mutant *Tmem98ΔN-FLAG* was generated by deleting the first 34 amino acids using an internal XhoI site. A slightly modified construct, in which a stretch of superfluous amino acids was removed by insertion of a HindIII/EcoRI oligomer behind the C-terminus of Tmem98, was used for most of the experiments.

A Tmem98-AcGFP fusion construct was generated by cloning the Tmem98 coding sequence into the SmaI cut pAc-GFP-N2 vector (Clontech). TMEM98-AcGFP did not inhibit FRAT2 activity to a similar extent as TMEM98-FLAG (data not shown). Because AcGFP has the tendency to dimerize [71], we later replaced this construct with *Tmem98-SGFP2*. Because GFP is not fluorescent at lower pH, we also generated *Tmem98-mTurquoise2* and *Tmem98-m-Cherry* fusions. This was achieved by exchanging *AcGFP1* with either *sGFP2, mTurquoise2 (mTq2)* or *mCherry* fragments using a BamHI/BsrGI digest.

*Tmem98* knockdown constructs were generated by annealing sense and antisense oligo's and cloning the annealed products into pRetrosuper (a gift from Dr. Thijn Brummelkamp, Netherlands Cancer Institute). Knockdown constructs recognize both mouse *Tmem98* and human *TMEM98* based on sequence homology. Sense oligo sequences: 5′-GATCCCCCTGGAAGCATGGAGACTGTTTCAAGAGAACAGTCTCCATGCTTCCAGTTTTTGGAAA-3′ (RNAi1), 5′-GATCCCCCCATCTTGAAGATTTGTCATTCAAGAGATGACAAATCTTCAAGATGGTTTTTGGAAA-3′ (RNAi2) and 5′-GATCCCCACATCATTGTGGTGGCCAATTCAAGAGATTGGCCACCACAATGATGTTTTTTGGAAA-3′ (RNAi3).

Additional constructs used were Lck-mCherry, Rab7-mCherry and Lamp1-mCherry (a gift from Dr. Joachim Goedhart, University of Amsterdam); Rab9a-mCherry (Addgene plasmid #78592, donating investigator Yihong Ye [72]), pLenti-CMV-rtTA3 Hygro (Addgene plasmid #26730, donating investigator Eric Campeau), pTREtight2 (Addgene plasmid #19407, donating investigator Markus Ralser), CMV-Renilla (Promega) and MegaTOPFLASH (a gift from Dr. Christophe Fuerer and Dr. Roel Nusse, Stanford University).

To create a stable doxycycline-inducible TMEM98-FLAG cell line, *Tmem98-FLAG* was amplified by PCR from *pCDNA3.1-Tmem98-FLAG* and inserted into pTREtight2 as an EcoRI/EcoRI fragment. All constructs were verified by restriction enzyme digestion analysis and Sanger sequencing prior to use.

The following constructs will be made available via Addgene: *pGlomyc-Frat1* (plasmid #124499), *pGlomyc-Frat2* (#124500), *pCDNA3.1-Tmem98-FLAG* (#124501), *pCDNA3.1-Tmem98ΔN-FLAG* (#124502), *Tmem98-sGFP2* (#124503), *Tmem98-mTq2* (#124504), *Tmem98-mCherry* (#124505) and *pTRE-tight2-Tmem98-FLAG* (#124504).

### Cell culture and transfection

HEK293TN (a gift from Dr. Anton Berns), HEK293A (a gift from Dr. Anton Berns) and HeLa cells (a gift from Dr. Joachim Goedhart) were grown in Dulbecco's modified Eagle's medium

supplemented with 10% fetal bovine serum and 1% penicillin/streptomycin (Gibco) under 5% $CO2$ at 37˚C in humidifying conditions. On the day prior to transfection, cells were plated in 6-well tissue culture plates, 12-well tissue culture plates, or 8-well chamber slides. Cells were transfected with a total amount of 200 ng DNA (per 8-chamber slide), 500 ng DNA (per well of a 12-well plate) or 1500 ng DNA (per well of a 6-well plate) using polyethylenimine (PEI, Polysciences Inc., dissolved at 1 mg/ml in ethanol). Transfection mixtures were made in Opti-mem using a 1:3 ratio (µg:µl) of DNA and PEI. In all cases, empty pGlomyc vector was added to control for the total amount of DNA transfected. Master mixes were made where possible to reduce variation.

To generate a stable doxycycline inducible TMEM98-FLAG cell line, HEK293TN cells were transfected with the *pTREtight2-Tmem98-FLAG* and *pLenti-CMV-rtTA3-Hygro* constructs. Following hygromycin selection, individual clones were picked and tested for TMEM98 induction by Western blot analysis. FACS analysis revealed that even within a clonal population of cells, not all cells induced TMEM98-FLAG expression. Subcloning did not solve this problem, suggesting that some cells had either lost or randomly silenced the construct.

Where indicated, cells were treated with MG132 (Sigma-Aldrich), bafilomcycin (Sigma-Aldrich) or Brefeldin A (Sigma Aldrich) for the times mentioned.

## Protein gels and Western blot analysis

HEK293TN cells transfected as described above with the indicated constructs were harvested 48 hours post-transfection by lysis in RIPA buffer supplemented with protease inhibitors (Roche) or in passive lysis buffer (Promega). Protein concentration was determined using a colorimetric assay (BioRad) or Pierce BCA protein assay (Thermo Scientific).

For immunoprecipitation, HEK293TN cells were transfected in a 6-well plate. Protein lysates were incubated at 4˚C with a mouse monoclonal antibody directed against the FLAG-tag (M2, Stratagene). Immunocomplexes were pulled down by incubation with protein G sepharose, after which samples were washed in RIPA buffer to remove unbound protein, resuspended in RIPA buffer, run on precast protein gels (either 10% or 4%-12%, Nupage) and analyzed on Western blot. Equal amounts of protein were prepared in passive lysis buffer with protein loading buffer (125 mM Tris-HCl (pH 6.8), 50% glycerol, 4% SDS, 0.2% Orange-G, 10% betamercaptoethanol) and samples were boiled at 95C prior to loading.

For the Western blots that gave rise to Fig 2G, Fig 4B and S7A–S7C Fig and S9 Fig, samples were run on 12% SDS-PAGE gel in electrophoresis buffer (25 mM Tris base, 192 mM Glycine, 0.1% SDS) at 80-120V and transferred to 0.2 µm nitrocellulose membrane (Biorad) overnight at 30V. Membranes were blocked in 1:1 TBS:Odyssey Blocking buffer (LI-COR) and incubated overnight at 4C with primary antibodies (rabbit anti-TMEM98, Proteintech, 1:1000; rabbit anti-FLAG polyclonal, Sigma, 1:2000; mouse anti-FLAG monoclonal M2, Stratagene 1:2000, mouse anti-myc monoclonal 9E10, Invitrogen, 1:5000; mouse anti-myc monoclonal 9B11, Cell Signaling Technologies, 1:1000; mouse anti-tubulin, Sigma-Aldrich, 1:1000; mouse anti-Hsp90a/b, mouse, Santa-Cruz, 1:1000, mouse anti-GSK3ß, BD Transduction labs, 1:2000) in TBS:Odyssey Blocking buffer supplemented with 0.1% Tween-20. Secondary antibody (anti-mouse 680, LI-COR, 1:20,000; anti-rabbit 800, LI-COR, 1:20,000) incubation was performed in TBS-T for 45 minutes at room temperature. Membranes were stored at 4C in TBS and imaged on an Odyssey Fc (LI-COR) for two minutes at 680 nm and 800 nm. These blots were used for quantification. For the Western blots depicted in Fig 1B and 1C, Fig 4C and 4D, S5 Fig and S6 Fig, samples were run on precast protein gels (either 10% or 4%-12%, Nupage) and blots were imaged using ECL detection (Pierce) instead, after labelling with goat-anti-mouse-HRP and goat-anti-rabbit-HRP secondary antibodies. These blots were not used for

quantification, since it could not be excluded that some of the signal on the film was oversaturated. Original Western blot files are available via the Open Science Framework at http://dx.doi.org/10.17605/OSF.IO/EF74W.

## Luciferase assay

For luciferase assay experiments, triplicate transfections were performed in 12-well plates. For each well, HEK293TN cells were transfected with 500 ng DNA total (usually 100 ng MegaTOP-FLASH, 50 ng CMV-Renilla and different amounts (25, 50, 100, 200 ng) of myc-Frat2 and Tmem98 constructs as required, supplemented with empty pGlomyc vector as carrier DNA. Cells were harvested 48 hours post-transfection in passive lysis buffer (Promega) and analysed with by dual luciferase assay in a Lumat LB 9507 Luminometer (Berthold Technologies) or a GloMax navigator (Promega). For each replicate, 10 µl of lysate was transferred to a black 96-well Optiplate using 50 µl of Firefly and 50 µl of Renilla detection reagents (reagents were either from Promega or home-made, according to a protocol shared by Dr. Christophe Fuerer). The ratio of Firefly:Renilla values was used as a measure of TOPFLASH activation. Within each experiment, all data were normalized to empty pGlomyc transfected cells, the Firefly:Renilla ratio of which was set to 1. Experiments were performed at least three times. For Western blot analyses of luciferase assay experiments, the lysates from triplicate wells were pooled after the luciferase assay measurements were performed.

## FACS analysis

A stable, doxycycline-inducible, clonal *tetO-Tmem98/CMV-rtTA3* HEK293TN cell line was treated with doxycycline (1 µg/ml) for 24 hours to induce TMEM98-FLAG expression. Vehicle treated cells were taken along as a negative control. Cells were then washed with PBS, trypsinized, and resuspended in HBSS/5%FBS. A small aliquot was taken as an unfixed, unstained control sample. For extracellular staining, unfixed cells were stained with an anti-FLAG M2 antibody (1:100, Sigma) on ice for 20 minutes, followed by labelling with a secondary antibody Donkey-anti-Mouse Alexa 488 (1:500, Molecular Probes) on ice for 20 minutes. For total intracellular and extracellular staining, cells were dissolved in 100 ul of Fixation/Permeabilization buffer A (BD Biosciences) and incubated at room temperature for 15 minutes. Following a wash with HBSS/5%FBS, cells were incubated with 100 ul Permeabilization buffer B (BD Biosciences) prior to staining with the anti-FLAG and Donkey/Mouse Alexa488 antibodies as described above. Cells were then washed in HBSS/5% FCS, dissolved in 400 ul HBSS/5% FBS and analysed on a BD FACS AriaIII. Note that we cannot exclude that trypsinization may have resulted in partial cleavage of the FLAG-tagged C-terminus of TMEM98, which may have resulted in an underestimation of the signal.

## Confocal microscopy

For immunofluorescence analysis, cells were plated in 8-well chamber slides and transfected on either the same day or the following day with a total amount of 200 ng (per 8-chamber slide) DNA using PEI. For direct fluorescent detection of fluorescent fusion proteins, cells were seeded onto glass coverslips in a 6 well plate and transfected with a total amount of 500 ng DNA per well.

Cells were imaged at 48 hours after transfection. The experiments depicted in Fig 2A, Fig 6B and Fig 6C were performed on live cells in microscopy medium (20 mM HEPES, pH 7.4; 137 mM NaCl, 5.4 mM KCl, 1.8 mM CaCl2, 0.8 mM MgCl2, 20 mM glucose). Lysotracker red (Invitrogen) was added 5–15 minutes prior to analysis. For all other imaging experiments, cells were fixed in 4% paraformaldehyde or ice-cold methanol. For fluorescent protein

detection, cells were washed in PBS and mounted in Mowiol mounting medium. For immuno-fluorescence staining, cells were permeabilized with 0.2% Triton-X100 or 0.1% Saponin and stained with antibodies directed against the FLAG tag (M2, Sigma, 1:400), Golgin97 (CDF4, Molecular Probes, 1:100), EEA1 (BD Biosciences, 1:100–1:200) and MPR (1:100–1:200). Nuclei were counterstained with TOPRO3 or DAPI. Secondary antibodies were Alexa conjugated Goat-anti-Mouse, Donkey-anti-Mouse or Goat-anti-Rabbit antibodies with Alexa 488, 568 or 633 dyes. Samples were imaged by sequential scanning on a Leica SP2 or SP8, or a Nikon A1 confocal microscope, using 405nm, 457nm, 488nm, 561nm and 633nm lasers and appropriate filter blocks or AOBS settings.

## Software, in silico analysis and online repositories

Image studio Lite (LI-COR) was used for quantitative Western blot analysis. Luciferase experiments were analysed in Excel. Graphs were made in GraphPad Prism and R Studio. Statistical testing was performed in GraphPad Prism. FACS data were analysed with FlowJo software.

Confocal microscopy images were processed in Fiji. Overlays were made using the Image5D plugin, using green and magenta for dual channel overlays. For single channel images, appropriate LUTs were selected for contrast and visualization purposes. Figures were made in Adobe Illustrator.

Online topology prediction algorithms (HMMTOP; Phobius; TMHMM; TMpred; DAS-TMfilter, PrediSi, SignalP 4.1, WOLF PSORT, PredictProtein, SherLoc2, Secretome 2.0) were used to predict the TMEM98 topology (Tables 3 and 4).

Multiple sequence alignments were made at https://www.ebi.ac.uk using data extracted from the Ensembl genome database (https://www.ensembl.org) as input (Fig 1 and S1 and S2 Figs). Version 3.4 of the BioGRID protein-protein interaction database was accessed on 9 June 2018 at https://thebiogrid.org (S8 Fig). Evolutionary Conserved Regions in human TMEM98 (Fig 1 and S1 Fig) were determined using the Aminode Evolutionary Analysis tool available at http://www.aminode.org. [73]. The LIFEdb database [27–29] was accessed at https://www.dkfz.de/en/mga/Groups/LIFEdb-Database.html to find images depicting the subcellular localization of N-terminal and C-terminal TMEM98 fusions (S3 Fig).

## Supporting information

**S1 Fig. Evolutionary conservation of TMEM98.**
(TIF)

**S2 Fig. TMEM98 clones identified in the yeast two hybrid screen with FRAT2 and FRAT2ΔN.**
(TIF)

**S3 Fig. Improper trafficking of an N-terminal TMEM98 fusion protein.**
(TIF)

**S4 Fig. Gating controls for FACS analysis.**
(TIF)

**S5 Fig. Interaction between FRAT1, FRAT2 and TMEM98.**
(TIF)

**S6 Fig. Knock down of TMEM98.**
(TIF)

**S7 Fig. Non-linear interaction between TMEM98 and FRAT2.**
(TIF)

**S8 Fig. Putative interactors of TMEM98.**
(TIF)

**S9 Fig. TMEM98 is not degraded by the lysosome.**
(TIF)

## Acknowledgments

We thank Lenny Brocks and Lauran Oomen (Netherlands Cancer Institute) as well as Ronald Breedijk (University of Amsterdam) for assistance with confocal microscopy, Katrin Wiese for help with designing and performing the FACS experiment, Saskia de Man for critical reading of the manuscript, and former and current colleagues at the Netherlands Cancer Institute and the University of Amsterdam for reagents, discussions and suggestions for experiments. We thank Anton Berns, in whose lab this work was initiated, for critical feedback and support.

## Author Contributions

**Conceptualization:** Tanne van der Wal, Renée van Amerongen.

**Formal analysis:** Tanne van der Wal, Renée van Amerongen.

**Funding acquisition:** Renée van Amerongen.

**Investigation:** Tanne van der Wal, Jan-Paul Lambooij, Renée van Amerongen.

**Methodology:** Renée van Amerongen.

**Project administration:** Renée van Amerongen.

**Resources:** Renée van Amerongen.

**Supervision:** Renée van Amerongen.

**Visualization:** Tanne van der Wal, Renée van Amerongen.

**Writing – original draft:** Tanne van der Wal, Renée van Amerongen.

**Writing – review & editing:** Tanne van der Wal, Renée van Amerongen.

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
