## [Decision Letter · Decision Letter 0]

14 Aug 2019

PONE-D-19-18523

TMEM98 is a negative regulator of FRAT mediated Wnt/β-catenin signalling

PLOS ONE

Dear dr. van Amerongen,

Thank you for submitting your manuscript to PLOS ONE. After careful consideration, we feel that it has merit but does not fully meet PLOS ONE’s publication criteria as it currently stands. Therefore, we invite you to submit a revised version of the manuscript that addresses the points raised during the review process.

Both reviewers were enthusiastic about the study but raised significant concerns about the dataset.  Of particular importance are interpretations of the imaging data in Figure 2 and a need for additional statistical analysis, especially Figures 3-5.  Please ensure that you fully address all of the reviewers' concerns in your revised manuscript.

We would appreciate receiving your revised manuscript by Sep 28 2019 11:59PM. To enhance the reproducibility of your results, we recommend that if applicable you deposit your laboratory protocols in protocols.io, where a protocol can be assigned its own identifier (DOI) such that it can be cited independently in the future. For instructions see: http://journals.plos.org/plosone/s/submission-guidelines#loc-laboratory-protocols

We look forward to receiving your revised manuscript.

Kind regards,

Michael Koval

Academic Editor

PLOS ONE

Journal Requirements:

2. For reproducibility purposes please modify your methods section to include the source of all cell lines used in this work. If already published, please provide a reference, if obtained commercially or from a colleague, please provide the name of the company/colleague.

This work was funded by a MacGillavry fellowship from the University of Amsterdam (to RvA) and a grant from the Centre of Biomedical Genetics (CBG) to Anton Berns.

This work was funded by a MacGillavry fellowship from the University of Amsterdam (to RvA). The funders had no role in study design, data collection and analysis, decision to publish, or preparation of the manuscript.

Reviewers' comments:

Reviewer's Responses to Questions

**Comments to the Author**

1. Is the manuscript technically sound, and do the data support the conclusions?

Reviewer #1: Partly

Reviewer #2: Yes

2. Has the statistical analysis been performed appropriately and rigorously? 

Reviewer #1: No

Reviewer #2: Yes

3. Have the authors made all data underlying the findings in their manuscript fully available?

Reviewer #1: Yes

Reviewer #2: Yes

4. Is the manuscript presented in an intelligible fashion and written in standard English?

Reviewer #1: Yes

Reviewer #2: Yes

5. Review Comments to the Author

Reviewer #1: The FRAT family of proteins have been shown to be activators of Wnt/β-catenin signalling. They act by binding to GSK3 thereby interfering with the assembly of the complex that phosphorylates free CTNNB1 and targets it for degradation. However, FRAT proteins are not required for Wnt signalling in mammals and so they are thought to be modulators. In this paper the authors carry out a yeast 2-hybrid experiment in order to detect proteins that interact with FRAT2 and identify the transmembrane protein TMEM98 as a potential binding partner. They go on to confirm the interaction between the two proteins by co-immunoprecipitation. They then investigate the sub-cellular localisation and topology of TMEM98 and find that it is recycled between the plasma membrane and the Golgi. They also examine the effect of TMEM98 on FRAT activity and protein levels.

Whilst I find the yeast 2-hybrid, interaction data between FRAT2 and TMEM98, the topology data and some of the localisation data to be convincing I do not find some of the other data compelling enough to fully support the conclusions drawn by the authors as outline below.

1) It does seem strange that GSK3 was not found in the yeast 2-hybrid experiment, this is worthy of discussion.

2) In Westerns TMEM98-FLAG appears to be a double band (Figs 1, 2 and 4). It appears to be a single band in other Westerns, for example Supplementary Fig 5. Also there are no size markers shown for any of the Westerns. Can the authors comment on this?

3) In Fig 2A there is clear plasma membrane staining for the tagged TMEM98. Fig 2C shows localisation of TMEM98 to the outer periphery following release from Brefeldin A block. The arrowheads purport to indicate TMEM98 at the periphery 180 minutes after release. I do not find this particularly convincing, it is very hard to see on the printed picture and when enlarged on the screen I can see just a faint blue edge. This is nothing like the pattern of staining seen in Fig 2A. What does an untransfected cell look like? Is there no fluorescent signal?

4) Fig 2D. For the TMEM98ΔN-AcGFP transfection again all the cells look blue, is this low or no signal?

5) Fig2E. For the TM1-AcGFP transfection arrowheads are supposed to point to signal at distal protrusions of the ER. Again. very low signal and difficult to see. The authors describe the protein AcGFP as being cytosolic. However the intensity of the heat map suggests to me that the principal place the protein is found is in the nucleus not the cytoplasm.

6) In the TOPFLASH reporter assays shown in Fig 3 is the effect of TMEM98 on FRAT-induced reporter activity statistically significant? Although we should not be slaves to p values it would be informative to know what they were as there is considerable spread and overlap in the data points. The authors state that TMEM98 reduced the level of CTNNB1/TCF signalling in a dose-dependent manner (Fig 3A) but for the highest amount of TMEM98 the median level of signalling is higher than for all the lower amounts of TMEM98 so this statement is not justified.

7) The authors conclude that there is a dose-dependent reduction in both FRAT1 and FRAT2 levels caused by increasing amounts of TMEM98 (Fig 4A and Supplementary Fig 5C). By eye it is not convincing to me, the protein levels need to be quantitated to conclude this. The authors do some quantitation shown in Fig 4B and C but the amount of TMEM98-sGFP2 does not look to me as though it increases with increased amount of transfected plasmid. In Fig 4D the level of myc-FRAT1 appears much lower when transfected alone that when it was transfected along with TMEM98-FLAG (compare lanes 1 and 4). Also, there does not appear to be a difference between the signal for FRAT proteins in the presence or absence of full-length TMEM98 in Supplementary Fig 5A. In Supplementary Fig 7A increasing amounts of TMEM98-FLAG are transfected along with a constant amount of myc-FRAT2. However in lane 4 there appears to be less TMEM98-FLAG than in lane 3. In lane 4 the signal for myc-FRAT2 is lower than the adjacent lanes where the amount of TMEM98-FLAG detected is higher. Overall there is a lot of variability among the different experiments. The authors themselves say this on page 7 of the manuscript.

8) Statistics should be done to test the significance of the TOPFLASH reporter assays shown in Fig 4E and F. There is a great deal of overlap in the pink box (just TMEM98 RNAi) and the blue and green boxes (TMEM98 RNAi along with myc-FRAT1 or myc-FRAT2).

9) In Supplementary Fig 7B and C the amount of TMEM98-FLAG is kept constant and the amounts of myc-FRAT1 and myc-FRAT2 are increased. The authors conclude that increased myc-FRAT2 increased TMEM98-FLAG levels but to me the signal from lane 1 (TMEM98-FLAG alone) looks extremely similar to the signal in 3 out of 4 of the lanes where increasing amounts of myc-FRAT2 were transfected. It is difficult to reach a convincing conclusion as to the effect of these two proteins on each other’s stability.

10) What do the yellow and blue arrowheads indicate in Fig. 6?

11) The reference list is duplicated.

Reviewer #2: In this manuscript, van der Wal described that TMEM98 transmembrane protein interacts with FRAT2 and regulates its activity. The findings are interesting as they reveal a new layer of regulation of Wnt/b-catenin signaling. However, there are several major concerns that need to be addressed prior to publication.

1. Figure 1：The binding between TMEM98 and FRAT2 appears to be very weak, as judged by the ratio between the FRAT2 band to the background bands which are faint in aFLAG but extremely strong in a-myc staining. This should be explained and discussed.

2. In consideration of the false positive signal in yeast-two-hybrid assay, the interaction between TMEM98-ΔN and FRAT2 is not shown and should be repeated by co-immunoprecipitation in the presence of the proteasome inhibitor MG132, including Fig S5. It is necessary to show the binding between TMEM98 and FRAT2-ΔN as well.

3. Figure 2: The data in Fig 2C showed that TMEM98-GFP signals were found to be closely associated with the nucleus in the absence of treatment, suggesting its presence in ER as well. In the presence of Brefeldin A, the signals were restricted to the Golgi, does Brefeldin A disturb the export from distal Golgi compartments to the cell surface (Miller SG, 1992), and does Brefeldin A inhibit protein transport from the Golgi to the ER? If so, these data should suggest that TMEM98 is transported from Golgi to the ER? The extracellular location of TMEM98 C-terminus in Fig 2A was not well verified . Other methods should be performed to substantiate the subcellular localization of TMEM98, such as using the technique defined in (Lorenz H, 2006). And, the TMEM98ΔN-AcGFP was invisible (or the weak signal appeared as blue?) in Fig 2D.

4. Figure 2H: it is mentioned in methods that cells were treated with trypsin in FACS analysis. However, trypsin will remove the extracellular C-terminus of TMEM98-Flag. Did this affect the results?

5. Figure 3: since FRAT1 is a more potent activator of CTNNB1/TCF signaling than FRAT2 showed in Fig 3B, is it better to test the inhibition of FRAT1 activities in the presence of TMEM98 in Fig 3A? Meanwhile, the conclusion is not convincing that TMEM98 consistently reduced the level of CTNNB1/TCF signaling in a dose-dependent manner according to Fig 3A (maybe due to the non-linear increase in TMEM98 protein levels). Did the level of phosphorylated β-catenin proteins change when transfected with TMEM98 and FRAT1/2?

6. As TMEM98-AcGFP protein does not inhibit FRAT2 activity to the same extent as TMEM98-FLAG and TMEM98-mCherry, it may not be suitable for subcellular tracing TMEM98 proteins in Fig 2 and Fig 6.

7. If the model in Fig 7B is correct, we would expect that the region between two transmembrane domains would interact with FRAT. Is this true?

References:

Lorenz H,Hailey DW and Lippincott-Schwartz J. Fluorescence protease protection of GFP chimeras to reveal protein topology and subcellular localization. Nat Methods, 2006, 3(3): 205-210.

Miller SG,Carnell L and Moore HH. Post-Golgi membrane traffic: brefeldin A inhibits export from distal Golgi compartments to the cell surface but not recycling. 1992, 118(2): 267-283.

6. PLOS authors have the option to publish the peer review history of their article (what does this mean?). If published, this will include your full peer review and any attached files.

Reviewer #1: No

Reviewer #2: No

---

## [Author Response · Author response to Decision Letter 0]

10 Nov 2019

Response to specific editor comments:

As indicated in your decision letter (PONE-D-19-18523), we have paid particular attention to the following:

- For the presentation and interpretation of the imaging data in Figure 2, we have changed the LUT for some of the figure panels to provide better contrast. We have also modified some of the main text in our manuscript to prevent overinterpretation of our data.

- For Figures 3-5 we have performed statistical testing as requested. We have also included new Western blots with quantification, since both reviewers commented on the variability of the data.

Please note, that the overall interpretation of our data has not changed. Additional details and background are provided in the attached rebuttal letter. 

As for compliance with PLOS ONE journal requirements, we have ensured the following:

1. All of the files in our resubmission meet PLOS ONE style requirements. 

2. The sources of all cell lines have been indicated in the methods section.

3. All original uncropped and unadjusted images underlying the blot and gel results have been posted at a public data repository at https://osf.io/ef74w/ (DOI provided in the methods section). 

4. All instances of data not shown have been removed from the manuscript.

5. We have removed funding-related text from the Acknowledgements section. The funding statement as originally provided is still correct.

Response to specific reviewer comments (also detailed in the attached rebuttal):

We thank both Reviewer #1 and Reviewer #2 for their time and effort. Based on their constructive feedback, we have made the required changes to our manuscript. We are confident that we have addressed all of their concerns and that our manuscript has improved as a result.

Reviewer #1 

The FRAT family of proteins have been shown to be activators of Wnt/β-catenin signalling. They act by binding to GSK3 thereby interfering with the assembly of the complex that phosphorylates free CTNNB1 and targets it for degradation. However, FRAT proteins are not required for Wnt signalling in mammals and so they are thought to be modulators. In this paper the authors carry out a yeast 2-hybrid experiment in order to detect proteins that interact with FRAT2 and identify the transmembrane protein TMEM98 as a potential binding partner. They go on to confirm the interaction between the two proteins by co-immunoprecipitation. They then investigate the sub-cellular localisation and topology of TMEM98 and find that it is recycled between the plasma membrane and the Golgi. They also examine the effect of TMEM98 on FRAT activity and protein levels. Whilst I find the yeast 2-hybrid, interaction data between FRAT2 and TMEM98, the topology data and some of the localisation data to be convincing I do not find some of the other data compelling enough to fully support the conclusions drawn by the authors as outline below.

We thank Reviewer #1 for their critical reading and evaluation of our manuscript. We are happy that they find the yeast-two-hybrid data, FRAT2 and TMEM98 interaction data, TMEM98 topology data and most of the localization data convincing, since these data are the foundation for the core message of our manuscript. We also appreciate the reviewer’s detailed comments and questions regarding some of the other data, and we want to take the opportunity to address all of their points in more detail below. 

1) It does seem strange that GSK3 was not found in the yeast 2-hybrid experiment, this is worthy of discussion.

Like the reviewer, we were initially surprised that we did not pick up GSK3 in our yeast-two-hybrid screen with full-length FRAT2. We reasoned that perhaps the FRAT2 N-terminus was masking the GSK3-interacting domain under the conditions used for the screen. This was one of the main reasons for repeating the yeast-two-hybrid screen with the N-terminal deletion mutant (FRAT2�N). However, as the reviewer rightly notes, we did not identify GSK3 in the FRAT2�N screen either. 

The simplest explanation is that for whatever reason the correct GSK3 clone/functional GSK3 fragment was not present in the yeast-two-hybrid cDNA library used for our Y2H screens. Another reason could be that the interaction between FRAT2 and GSK3 is relatively weak and/or transient. We don’t have any actual data to support either one of these scenarios. However, the reviewer might be interested to know, that the lab of Dr. Anton Berns (where this work was initiated) also never picked up GSK3 in any of the FRAT1 yeast-two-hybrid screens that were performed in the mid 1990s (this work remains unpublished).

Since we do not have a clear explanation as to why GSK3 did not show up as an interactor in our yeast-two-hybrid screen (as noted in our revised manuscript on page 5, lines 91-92), we chose not to spend too much space in the manuscript speculating about the potential reasons. 

2) In Westerns TMEM98-FLAG appears to be a double band (Figs 1, 2 and 4). It appears to be a single band in other Westerns, for example Supplementary Fig 5. Also there are no size markers shown for any of the Westerns. Can the authors comment on this?

We apologize for the absence of the protein size markers. This was an omission on our part, which has now been corrected in the revised version of the manuscript. In addition, all of the original, uncropped blots have now also been made available via https://osf.io/ef74w/ (as per PLOS ONE guidelines). 

As for why TMEM98-FLAG appears as a double band on some of our blots but not on others: We hypothesize that the two bands represent different post-translationally modified species of TMEM98, but we have not investigated this further as this was beyond the scope of the current study. 

We suspect that the double bands are not visible on all of our blots because not all experiments were performed under the exact same conditions. For instance, some Western blots contain protein lysates made with RIPA lysis buffer that were run on a 4-12% NuPage pre-cast gel, whereas others depict protein lysates made with Promega’s passive lysis buffer (e.g. when these samples were also used for luciferase assays) that were run on home-made 12% SDS-page gels. These differences are detailed in the materials and methods section (revised manuscript, pages 29-30). 

3) In Fig 2A there is clear plasma membrane staining for the tagged TMEM98. Fig 2C shows localisation of TMEM98 to the outer periphery following release from Brefeldin A block. The arrowheads purport to indicate TMEM98 at the periphery 180 minutes after release. I do not find this particularly convincing, it is very hard to see on the printed picture and when enlarged on the screen I can see just a faint blue edge. This is nothing like the pattern of staining seen in Fig 2A. What does an untransfected cell look like? Is there no fluorescent signal?

We agree with the reviewer that plasma membrane staining is more clear in some experiments than in others. We can offer multiple explanations for this: First, the experiments were performed in different cell lines (e.g. HeLa versus HEK293A) and with different fixations (e.g. live-cell imaging of unfixed cells, versus 4% PFA or methanol fixation). We observed the clearest plasma membrane staining in living, unfixed HeLa cells (hence the example in Figure 2A). Second, the most prominent plasma membrane signal is not always detected in the same plane (or Z-position) as the ER or the Golgi signal (see Figure 2B as an example, where the focus is on the Golgi rather than the plasma membrane; indicated in the Figure legend (revised manuscript, page 10, lines 183-188). Third, the reviewer is correct in noting that the plasma membrane signal is relatively faint compared to other subcellular structures (including the Golgi), particularly in fixed cells (e.g. Figure 2C). In these images, we set our PMT/gain so as not to oversaturate the most intense signal (Golgi/ER), causing the plasma membrane signal to be relatively low.

In our revised manuscript, we have changed the LUT for Figures 2C-E to improve the contrast/visualization. We think that as a result, the plasma membrane signal (indicated by arrowheads) is now better visible and we hope the reviewer agrees. 

4) Fig 2D. For the TMEM98ΔN-AcGFP transfection again all the cells look blue, is this low or no signal?

A similar point was raised by Reviewer #2. We include the image below for the reviewers, which also contains the negative control sample for the experiment depicted in Figure 2D. In the image below, we have enhanced the contrast and brightness such that the untransfected control cells (3rd panel) are just barely visible. Thus, the signal for TMEM98�N (2nd panel, as depicted in Figure 2D) is indeed low signal. 

When the contrast and brightness of the TMEM98�N transfected cells are adjusted further (4th panel, ‘boosted’), the untransfected cells in the well also become visible. In our revision, we have indicated the position of some of these cells with an asterisk in Figure 2D (see also figure legend, revised manuscript, page 10 lines 195-200). In addition, we have changed the LUT for Figures 2C-E for better contrast/visualization. Thus, we hope that the reviewer agrees that the cells transfected with TMEM98�N are indeed expressing low levels of the deletion mutant, which is in agreement with our Western blots result presented in Figure 1.

5) Fig2E. For the TM1-AcGFP transfection arrowheads are supposed to point to signal at distal protrusions of the ER. Again. very low signal and difficult to see. 

Again, we agree with the reviewer that the signal is subtle. In our revised manuscript, we have changed the LUT for Figures 2C-E to improve the contrast/visualization. Because we do not want to over interpret our data, we have removed the specific statement that signal reflects distal protrusions of the ER from the legend of Figure 2 (revised manuscript, pages 10-11, lines 200-203). We have also removed the arrowheads from panel 2E accordingly.

The authors describe the protein AcGFP as being cytosolic. However the intensity of the heat map suggests to me that the principal place the protein is found is in the nucleus not the cytoplasm.

We agree with the reviewer that in the image depicted, the signal is more intense in the nucleus. We have reworded the figure legend to reflect that AcGFP (when not fused to TMEM98) is a free floating fluorescent protein (revised manuscript, page 11, lines 203-205). 

We want to point out that it is not uncommon for free floating fluorescent proteins to be distributed across the nucleus and the cytoplasm as a result of diffusion (possible due to their small size). This may well result in the signal appearing to be more intense in the nucleus than in the cytoplasm. For a reference, see Seibel et al. 2007 (https://www.ncbi.nlm.nih.gov/pubmed/17586454).

6) In the TOPFLASH reporter assays shown in Fig 3 is the effect of TMEM98 on FRAT-induced reporter activity statistically significant? Although we should not be slaves to p values it would be informative to know what they were as there is considerable spread and overlap in the data points. 

We have performed statistical testing for Figures 3-5 as requested and have updated the figures, figure legends and materials and methods sections accordingly. As can be seen in Figure 3, the inhibitory effect of TMEM98 on FRAT induced reporter activity is statistically significant. 

The authors state that TMEM98 reduced the level of CTNNB1/TCF signalling in a dose-dependent manner (Fig 3A) but for the highest amount of TMEM98 the median level of signalling is higher than for all the lower amounts of TMEM98 so this statement is not justified.

We thank the reviewer for pointing this out. A similar point was raised by Reviewer #2. Given that we only had two data points for this particular condition (i.e. the highest amount of Tmem98), this precludes meaningful statistical testing. Therefore, we have removed this condition from Figure 3A. We have modified the text so as not to overinterpret our data and have removed all instances of “dose-dependent” inhibition. Page 13, lines 263-264 now reads: “Co-transfection experiments showed that TMEM98 reduced the level of CTNNB1/TCF signalling induced by FRAT2 (Fig 3A).” See also point 7 below.

7) The authors conclude that there is a dose-dependent reduction in both FRAT1 and FRAT2 levels caused by increasing amounts of TMEM98 (Fig 4A and Supplementary Fig 5C). By eye it is not convincing to me, the protein levels need to be quantitated to conclude this. The authors do some quantitation shown in Fig 4B and C but the amount of TMEM98-sGFP2 does not look to me as though it increases with increased amount of transfected plasmid. In Fig 4D the level of myc-FRAT1 appears much lower when transfected alone that when it was transfected along with TMEM98-FLAG (compare lanes 1 and 4). Also, there does not appear to be a difference between the signal for FRAT proteins in the presence or absence of full-length TMEM98 in Supplementary Fig 5A. In Supplementary Fig 7A increasing amounts of TMEM98-FLAG are transfected along with a constant amount of myc-FRAT2. However in lane 4 there appears to be less TMEM98-FLAG than in lane 3. In lane 4 the signal for myc-FRAT2 is lower than the adjacent lanes where the amount of TMEM98-FLAG detected is higher. Overall there is a lot of variability among the different experiments. The authors themselves say this on page 7 of the manuscript.

We appreciate the critical eye of this reviewer. 

First, the reviewer is correct in noting that “the amount of TMEM98-sGFP2 does not look to me as though it increases with increased amount of transfected plasmid”. This is indeed the case when TMEM98 is co-transfected with FRAT2 (see new Figure 4B and new Supplementary Figure 7 for additional examples). This observation is the reason for proposing the model in new Supplementary Figure 7E (formerly Figure 5B). Given that we don’t have definitive proof for this model, we have moved the model from the main figures to the supplementary figures. Accordingly, we moved this section from the results to the discussion, which now reads (revised manuscript, page 21, lines 439-444): 

“Because TMEM98 protein levels did not increase linearly in the presence of FRAT2, as opposed to when TMEM98 was transfected alone (Fig 4B and S7A-D Fig), our findings are consistent with a model in which TMEM98 and FRAT2 form a negative feedback loop (S7E Fig). As such, TMEM98 and FRAT2 may constitute a dynamic regulatory switch with the capacity to fine-tune CTNNB1/TCF signalling activity.”

Second, the reviewer is also correct in noting that “Overall there is a lot of variability among the different experiments. The authors themselves say this on page 7 of the manuscript.”

We initially suspected that these were pipetting errors, but after careful optimization of our experiments we suspect that this is actual, biological variation. We reported this variability to be transparent, but we now realize that it may have distracted from the main message of our manuscript, which is that TMEM98 binds and inhibits FRAT2. 

In our revised manuscript, we have modified Figure 4 and the corresponding main text to 

i) properly illustrate the inherent spread in our data, ii) allow proper quantification of the results and iii) perform statistical testing:

- We have removed the data in which we compare the effects of TMEM98 overexpression on FRAT1 and FRAT2 protein levels, so as not to confuse the reader or dilute our main message (which focuses on FRAT2). The text has been modified accordingly. The discussion (revised manuscript, page 21, lines 444-446) now reads: “TMEM98 also binds and inhibits FRAT1, but whether TMEM98 affects the biological activity of both homologues to the same extent remains to be determined.”

- We have included additional Western blots (for a total of 9 independent experiments) to quantify the effects of TMEM98 overexpression on FRAT2 protein levels (new Figure 4A, which now also includes statistical testing). The corresponding text (revised manuscript, page 15, lines 305-307) now reads: “Quantitative Western blot analysis revealed a variable, but consistent reduction in FRAT2 protein levels in the presence of full length TMEM98 (Fig 4A,B and S7A-C Fig).” 

- Some of the Western blots that were used in the first version of our manuscript (including those for the original Figure 4A and Supplementary Figure 5 and Supplementary Figure 7) may have been overexposed (as a result of using ECL detection and manual exposure of X-Ray film), thus preventing accurate/meaningful quantification. We have replaced these Western blots with new, non-saturated Western blots from new experiments using fluorescent antibodies and LICOR detection (new Supplementary Figure 7A-C). Although the corresponding results section (revised manuscript, page 15-16, lines 301-324), has been rewritten to focus on the TMEM98 mediated reduction of FRAT protein levels we want to stress that none of the changes we have made changes our original message or interpretation of the data.

We believe that as a result of these changes the flow of our story has improved and more speculative (e.g. the negative feedback loop model) and potentially confusing aspects (e.g. inherent variability) have been moved to the discussion and supplementary data. 

8) Statistics should be done to test the significance of the TOPFLASH reporter assays shown in Fig 4E and F. There is a great deal of overlap in the pink box (just TMEM98 RNAi) and the blue and green boxes (TMEM98 RNAi along with myc-FRAT1 or myc-FRAT2).

We have performed statistical testing for Figures 3-5 as requested and have updated the figures, figure legends and materials and methods accordingly. As can be seen in Figure 4E-F, only the increase in FRAT1 induced TOPFLASH reporter activity is statistically significant. We suspect that this is due to the fact that FRAT1 is a more potent activator of CTNNB1/TCF signaling than FRAT2 to begin with (see van Amerongen et al., 2004, JBC and van Amerongen et al., 2010, Oncogene), which allows for a larger dynamic range.

9) In Supplementary Fig 7B and C the amount of TMEM98-FLAG is kept constant and the amounts of myc-FRAT1 and myc-FRAT2 are increased. The authors conclude that increased myc-FRAT2 increased TMEM98-FLAG levels but to me the signal from lane 1 (TMEM98-FLAG alone) looks extremely similar to the signal in 3 out of 4 of the lanes where increasing amounts of myc-FRAT2 were transfected. It is difficult to reach a convincing conclusion as to the effect of these two proteins on each other’s stability.

Once again, we thank the reviewer for their careful consideration of our data and we agree that it is difficult to reach a convincing conclusion as to the effect of TMEM98 and FRAT2 on each other’s stability. As detailed in our response to point 7) above, we have moved this more speculative aspect to the discussion and we have rewritten the results section to improve the flow and not distract from our main message. 

10) What do the yellow and blue arrowheads indicate in Fig. 6?

We apologize for not including this in the figure legend.

Yellow arrowheads indicate co-localization, whereas blue arrowheads indicate examples without co-localization. This has been corrected in the figure legend (now Figure 5 in the revised manuscript, page 19, lines 397-398). 

11) The reference list is duplicated.

We apologize for this error. This has been fixed in the revised manuscript.

Reviewer #2

In this manuscript, van der Wal described that TMEM98 transmembrane protein interacts with FRAT2 and regulates its activity. The findings are interesting as they reveal a new layer of regulation of Wnt/b-catenin signaling. However, there are several major concerns that need to be addressed prior to publication.

We thank Reviewer #2 for their comments and constructive feedback. We are confident that we have addressed all of their concerns in the revised version of our manuscript. Below, we will address all of their points in more detail.

1. Figure 1：The binding between TMEM98 and FRAT2 appears to be very weak, as judged by the ratio between the FRAT2 band to the background bands which are faint in aFLAG but extremely strong in a-myc staining. This should be explained and discussed.

The reviewer is correct in noting that not all of the FRAT2 protein pool is bound by TMEM98. We agree that this can be explained by low affinity binding. However, we also offer another explanation: If our model, depicted in new Figure 6A (formerly Figure 7A), is correct, then only a small portion of the FRAT2 protein pool can come into contact with TMEM98. Thus, we think our immunoprecipitation data lend further support for a model in which the only interaction between FRAT2 and TMEM98 can take place inside MVBs. This has now been included in the discussion (page 22-23, lines 482-483: “our immunoprecipitation experiments support the fact that only a small proportion of the total FRAT2 protein pool interacts with TMEM98 (Fig 1).”). We thank reviewer 2 for pointing out this fact.

2. In consideration of the false positive signal in yeast-two-hybrid assay, the interaction between TMEM98-ΔN and FRAT2 is not shown and should be repeated by co-immunoprecipitation in the presence of the proteasome inhibitor MG132, including Fig S5. It is necessary to show the binding between TMEM98 and FRAT2-ΔN as well.

We have not been able to detect an interaction between TMEM98�N and FRAT2 or between TMEM98 and FRAT2�N in co-immunoprecipitation experiments. We suspect that this is at least in part due to the fact that both deletion mutants (TMEM98�N and FRAT2�N) are mislocalized. As an example, we provide Figure 2 for the reviewer below, which illustrates the tendency of FRAT2�N to accumulate in the nucleus (a nuclear export signal is known to reside in the FRAT N-terminus: Franca-Koh et al., 2002, JBC). A precise dissection of the interaction between TMEM98 and FRAT2 would thus require an entirely different experimental approach, and we argue that this is beyond the scope of the current manuscript.

However, we want to point out that we provide multiple lines of evidence that support a physical and functional interaction between TMEM98 and FRAT2, including co-immunoprecipitation of the full length proteins (Figure 1), the fact that TMEM98�N is stabilized when FRAT2 is co-expressed (Figure 1), and the effect of both TMEM98 overexpression and knockdown on FRAT1 and FRAT2 protein levels and activity (Figures 3 and 4).

3. Figure 2: The data in Fig 2C showed that TMEM98-GFP signals were found to be closely associated with the nucleus in the absence of treatment, suggesting its presence in ER as well. In the presence of Brefeldin A, the signals were restricted to the Golgi, does Brefeldin A disturb the export from distal Golgi compartments to the cell surface (Miller SG, 1992), and does Brefeldin A inhibit protein transport from the Golgi to the ER? If so, these data should suggest that TMEM98 is transported from Golgi to the ER? 

We suspect that the reviewer has perhaps misinterpreted Figure 2C and we apologize for any confusion we may have caused. As the legend indicates, the “0 minutes” timepoint in Figure 2C refers to the chase time after release from a 4-hour treatment with Brefeldin A (revised manuscript, page 10, lines 190-192: “Cells were treated with Brefeldin A for 4 hours to block forward trafficking through the Golgi, after which they were released. Cells were fixed at 0, 30, 60 and 180 minutes following release.”). Thus, the “0 minutes” timepoint does not reflect the absence of treatment, but the result of 4 hours of treatment with Brefeldin A. The reviewer is correct in noting that at this point, most of the TMEM98 signal is indeed restricted to the ER. This is in agreement with previous reports demonstrating a role for TMEM98 in this compartment (Huang et al., 2018, J Neuroscience). After a longer chase (60 minutes and longer after release from the Brefeldin A treatment) TMEM98 again traffics to the Golgi and the cell surface (see arrowheads in the 180 minute chase panel). 

The extracellular location of TMEM98 C-terminus in Fig 2A was not well verified . Other methods should be performed to substantiate the subcellular localization of TMEM98, such as using the technique defined in (Lorenz H, 2006). 

We thank the reviewer for suggesting that we perform a fluorescence protease protection (FPP) assay. We have considered this option to determine the orientation of TMEM98 in the plasma membrane. However, it would require us to also tag TMEM98 on the N-terminus and this is not possible due the presence of the N-terminal signal peptide and membrane anchor, which results in improper targeting and localization (as also shown in Supplementary Figure 3). This is one of the reasons why we chose to perform the FACS experiment depicted in Figure 2 instead. To our opinion, this experiment provides sufficient evidence that the C-terminus is located on the outside of the cell, confirming the experiments of Fu et al. (Fu et al., 2015, J Interferon Cytokine Res). 

And, the TMEM98ΔN-AcGFP was invisible (or the weak signal appeared as blue?)

in Fig 2D.

A similar point was raised by Reviewer #1. We include the image below for the reviewers, which also contains the negative control sample for the experiment depicted in Figure 2D. In the image below, we have enhanced the contrast and brightness such that the untransfected control cells (3rd panel) are just barely visible. Thus, the signal for TMEM98�N (2nd panel, as depicted in Figure 2D) is indeed low signal. 

When the contrast and brightness of the TMEM98�N transfected cells are adjusted further (4th panel, ‘boosted’), the untransfected cells in the well also become visible. In our revision, we have indicated the position of some of these cells with an asterisk in Figure 2D (see also figure legend, revised manuscript, page 10 lines 195-200). In addition, we have changed the LUT for Figures 2C-E for better contrast/visualization. Thus, we hope that the reviewer agrees that the cells transfected with TMEM98�N are indeed expressing low levels of the deletion mutant, which is in agreement with our Western blots result presented in Figure 1.

4. Figure 2H: it is mentioned in methods that cells were treated with trypsin in FACS analysis. However, trypsin will remove the extracellular C-terminus of TMEM98-Flag. Did this affect the results?

We thank the reviewer for pointing this out. We cannot exclude that indeed some signal may have been lost due to cleavage by trypsin. In fact, we believe that this might explain why less than 100% of the cell population shows up as positive in the extracellular staining. We have included a remark to this effect in the materials and methods section (revised manuscript, page 32, lines 694-696: “Note that we cannot exclude that trypsinization may have resulted in partial cleavage of the FLAG-tagged C-terminus of TMEM98, which may have resulted in an underestimation of the signal.”). However, we stress that this does not affect our main conclusion, which is that the C-terminus of TMEM98 is on the outside of the cell.

5. Figure 3: since FRAT1 is a more potent activator of CTNNB1/TCF signaling than FRAT2 showed in Fig 3B, is it better to test the inhibition of FRAT1 activities in the presence of TMEM98 in Fig 3A? 

We have performed our most elaborate set of experiments with FRAT2, given that this is how we initially identified TMEM98. We do find that FRAT1 also binds to TMEM98 (Supplementary Figure 5) and indeed, knockdown of TMEM98 increases FRAT1 mediated CTNNB1/TCF signaling (Figure 3). Thus, we propose that TMEM98 biologically interacts with both FRAT homologues. However, it is beyond the scope of the current study to investigate how similar and/or different the activities of TMEM98 towards FRAT1 and FRAT2 are. 

We have removed the data in which we compare the effects of TMEM98 overexpression on FRAT1 and FRAT2 protein levels, so as not to confuse the reader or dilute our main message (which focuses on FRAT2). The text has been modified accordingly. This is also mentioned in the discussion (revised manuscript, page 21, lines 444-446): “TMEM98 also binds and inhibits FRAT1, but whether TMEM98 affects the biological activity of both homologues to the same extent remains to be determined.”

Meanwhile, the conclusion is not convincing that TMEM98 consistently reduced the level of CTNNB1/TCF signaling in a dose-dependent manner according to Fig 3A (maybe due to the non-linear increase in TMEM98 protein levels). 

We thank the reviewer for pointing this out. A similar point was raised by Reviewer #1. Given that we only had two data points for this particular condition (i.e. the highest amount of Tmem98), this precludes meaningful statistical testing. Therefore, we have removed this condition from Figure 3A. We have modified the text so as not to overinterpret our data and have removed all instances of “dose-dependent” inhibition. Page 13, lines 263-264 now reads: “Co-transfection experiments showed that TMEM98 reduced the level of CTNNB1/TCF signalling induced by FRAT2 (Fig 3A).”

In our revised manuscript, we have modified Figure 4 and the corresponding main text to 

i) properly illustrate the inherent spread in our data, ii) allow proper quantification of the results and iii) perform statistical testing.

We have included additional Western blots (for a total of 9 independent experiments) to quantify the effects of TMEM98 overexpression on FRAT2 protein levels (new Figure 4A, which now also includes statistical testing). The corresponding text (revised manuscript, page 15, lines 305-307) now reads: “Quantitative Western blot analysis revealed a variable, but consistent reduction in FRAT2 protein levels in the presence of full length TMEM98 (Fig 4A,B and S7A-C Fig).” 

Some of the Western blots that were used in the first version of our manuscript (including those for the original Figure 4A and Supplementary Figure 5 and Supplementary Figure 7) may have been overexposed (as a result of using ECL detection and manual exposure of X-Ray film), thus preventing accurate/meaningful quantification. We have replaced these Western blots with new, non-saturated Western blots from new experiments using fluorescent antibodies and LICOR detection (new Supplementary Figure 7A-C). Although the corresponding results section (revised manuscript, page 15-16, lines 301-324), has been rewritten to focus on the TMEM98 mediated reduction of FRAT protein levels we want to stress that none of the changes we have made changes our original message or interpretation of the data.

Did the level of phosphorylated β-catenin proteins change when transfected with TMEM98 and FRAT1/2?

Since TMEM98 reduces FRAT2 protein levels, it is expected that the levels of non-phosphorylated or active beta-catenin indeed change accordingly. However, this is likely to be a secondary effect, due to the change if FRAT2 levels and therefore we have not investigated this further.

6. As TMEM98-AcGFP protein does not inhibit FRAT2 activity to the same extent as TMEM98-FLAG and TMEM98-mCherry, it may not be suitable for subcellular tracing TMEM98 proteins in Fig 2 and Fig 6.

Cranfill et al. reported the tendency of AcGFP to dimerize (Cranfill et al., 2016, Nature Methods), which is why we re-cloned our existing constructs into sGFP2 and mTq2 fusion proteins for our co-localization studies as a pre-caution. However, there is no reason to assume that an AcGFP fusion protein would result in major differences in trafficking at low levels of protein expression. In support of this, the plasma membrane and Golgi localization of TMEM98-AcGFP are confirmed by our results with TMEM98-sGFP2 (Figure 2A), as are other subcellular localization patterns in Figure 5 (formerly Figure 6), which were confirmed with co-localization of TMEM98-sGFP2 and/or TMEM98-mTq2 with Rab-fusion proteins (including Rab5, which supports our results for EEA1, data not shown). 

7. If the model in Fig 7B is correct, we would expect that the region between two transmembrane domains would interact with FRAT. Is this true?

The reviewer is correct that this is indeed our hypothesis. However, as we also point out, this would be hard to reconcile with some of the topology findings by Fu et al. (2015). At this point, both of our models are speculative. Future studies would be needed to determine the exact FRAT/TMEM98 binding interface. We have rewritten the discussion around Figure 6 (formerly Figure 7) to stress this point (revised manuscript, pages 21-23, lines 448-493).

---

## [Decision Letter · Decision Letter 1]

12 Dec 2019

PONE-D-19-18523R1

TMEM98 is a negative regulator of FRAT mediated Wnt/β-catenin signalling

PLOS ONE

Dear dr. van Amerongen,

Thank you for submitting your manuscript to PLOS ONE. After careful consideration, we feel that it has merit but does not fully meet PLOS ONE’s publication criteria as it currently stands. Therefore, we invite you to submit a revised version of the manuscript that addresses the points raised during the review process.

Both reviewers agree that the manuscript is considerably improved. However there remain some concerns that need to be addressed. Please ensure that all the statistics are complete and all figure legends are included in the manuscript.  Also, issues related to markers for immunoblots and the signal strength of plasma membrane labeling also need to be addressed.

We would appreciate receiving your revised manuscript by Jan 26 2020 11:59PM. To enhance the reproducibility of your results, we recommend that if applicable you deposit your laboratory protocols in protocols.io, where a protocol can be assigned its own identifier (DOI) such that it can be cited independently in the future. For instructions see: http://journals.plos.org/plosone/s/submission-guidelines#loc-laboratory-protocols

We look forward to receiving your revised manuscript.

Kind regards,

Michael Koval

Academic Editor

PLOS ONE

Reviewers' comments:

Reviewer's Responses to Questions

**Comments to the Author**

1. If the authors have adequately addressed your comments raised in a previous round of review and you feel that this manuscript is now acceptable for publication, you may indicate that here to bypass the “Comments to the Author” section, enter your conflict of interest statement in the “Confidential to Editor” section, and submit your "Accept" recommendation.

Reviewer #1: (No Response)

Reviewer #2: All comments have been addressed

2. Is the manuscript technically sound, and do the data support the conclusions?

Reviewer #1: Yes

Reviewer #2: Yes

3. Has the statistical analysis been performed appropriately and rigorously? 

Reviewer #1: Yes

Reviewer #2: Yes

4. Have the authors made all data underlying the findings in their manuscript fully available?

Reviewer #1: Yes

Reviewer #2: Yes

5. Is the manuscript presented in an intelligible fashion and written in standard English?

Reviewer #1: Yes

Reviewer #2: Yes

6. Review Comments to the Author

Reviewer #1: The paper is much improved. I just have a few minor comments to make.

Size markers are still not shown for all the Westerns, for example Fig 1B.

The authors say “In our revised manuscript, we have changed the LUT for Figures 2C-E to improve the contrast/visualization. We think that as a result, the plasma membrane signal (indicated by arrowheads) is now better visible and we hope the reviewer agrees.” To me the signal at the cell periphery is still quite faint.

In figure 3B please show the P value for the luciferase activity of myc-FRAT2 with and without TMEM98-FLAG.

Please show the P value for the luciferase activity of myc-FRAT2 with and without TMEM98 RNAi #3 in Fig 4E. I am not convinced Fig 4F adds anything. Is it just a way of showing the data in Fig 4E in a different way?

Apologies but I could not find the figure legends for the supplementary figures in the revised submission.

Reviewer #2: The reviewers have adequately addressed my questions and concerns, and explained why some experiments can not be done at this stage.

7. PLOS authors have the option to publish the peer review history of their article (what does this mean?). If published, this will include your full peer review and any attached files.

Reviewer #1: No

Reviewer #2: No

---

## [Author Response · Author response to Decision Letter 1]

16 Dec 2019

Please see our official cover letter for a direct response to the points highlighted by the editor. Our response to the reviewers is also attached as a separate rebuttal, and pasted below:

We thank both reviewers for assessing the revised version of our manuscript. Below we want to take the opportunity to address the remaining comments of Reviewer #1.

Reviewer #1: 

The paper is much improved. I just have a few minor comments to make.

Size markers are still not shown for all the Westerns, for example Fig 1B.

As shown in Figure 1, we have used the heavy chain (~50 kDa) and light chain (~25 kDa) bands from the antibodies used for the immunoprecipitation experiments as size markers (the same holds for Figure S5). Given that we had 12 samples to run for these pull downs on a 12-slot gel, this precluded the inclusion of a protein size marker. Therefore, we gave preference to running the lysates and the pull downs on a single gel over the inclusion of a separate marker. 

As shown in the original data deposited at DOI 10.17605/OSF.IO/EF74W, these Western blots are clean and the heavy and light chain can be used reliably to get an impression of protein size distribution on the blot.

Please note that the panel on the left comes from the same gel as the panel on the right (only the order has been inversed, to depict the total lysates on the left and the pull downs on the right). These markers thus refer to both panels (direct lysates and pull downs). 

We have double-checked all other Western blots in the main figures and supplementary figures, and all of them now have size markers. We apologize for not catching all of these omissions in our previous resubmission. 

Please note that the sole exception is the experiment depicted in Figure 4C/D, where we unfortunately forgot to copy the size markers onto the film after developing the Western blot (ECL detection). Here, we ask the reviewer to please also assess all of the original data deposited at DOI 10.17605/OSF.IO/EF74W, which shows that myc-Frat1/2 and Tmem98-FLAG reliably run at the expected sizes in all experiments (i.e. between 25 kDa and 35 kDa). 

The authors say “In our revised manuscript, we have changed the LUT for Figures 2C-E to improve the contrast/visualization. We think that as a result, the plasma membrane signal (indicated by arrowheads) is now better visible and we hope the reviewer agrees.” To me the signal at the cell periphery is still quite faint.

We agree with this reviewer that the signal in the plasma membrane is “still quite faint” for the images depicted in Figure 2C-E. However, we want to emphasize that our message is not that TMEM98 is predominantly located in the plasma membrane, but that it is present in the plasma membrane at all. 

As described on pages 8-9, TMEM98 is both predicted to be secreted as well as to be a transmembrane protein. Our experiments (mainly Figure 2A and Figure 2H) support that at least part of the TMEM98 pool resides in the plasma membrane, which we believe is an important confirmation of the resident location of TMEM98. Figure 2C supports this conclusion, since it shows that after being trapped in the ER following Brefeldin A treatment, part of the TMEM98 pool again reaches the plasma membrane 3 hours after release (arrowheads in Figure 2C). 

To show the reviewer additional examples, we provide the images below. These also show TMEM98 returning to the plasma membrane 2 hours (top left) and 3 hours (following 3 images) after release from Brefeldin A treatment. Please note that the image depicted in 2C is also shown in the images below (bottom right). 

2 hours after release. Note plasma membrane signal at the cell-cell border. 

3 hours after release. Note the subtle outline of the plasma membrane. 

3 hours after release. Note the subtle outline of the plasma membrane. 

3 hours after release (same as image 2C). Note the subtle outline of the plasma membrane and the visibility of the distal cell membrane protrusions.

The grey scale enhances the contrast, but the reviewer is right: the signal in the plasma membrane is subtle. As we already indicated in our previous rebuttal: We observed the clearest plasma membrane staining in living, unfixed HeLa cells (hence the example in Figure 2A). Please note that Figure 2C was made using PFA fixed cells, which in our hands reduced the intensity of the plasma membrane signal. The plasma membrane signal is also faint compared to other subcellular structures (including the Golgi, see also examples provided above), particularly in fixed cells (as e.g. Figure 2C). In these images, we set our PMT/gain so as not to oversaturate the most intense signal (Golgi/ER), causing the plasma membrane signal to be relatively low. Please note that all of these subtleties are indicated in the Legend for Figure 2.

Taken all of this together, we stand by our conclusion as stated in the manuscript on pages 8-9: “A TMEM98-GFP fusion protein indeed localizes to the plasma membrane (Fig 2A), in addition to showing more prominent localization to the Golgi and intracellular vesicles (Fig 2A-B).”

In figure 3B please show the P value for the luciferase activity of myc-FRAT2 with and without TMEM98-FLAG.

We analyzed our data with Prism, using a one-way ANOVA and Tukey’s multiple comparisons test (comparing every condition to every other condition). In this analysis, neither the difference between vector and myc-Frat2 alone (first condition vs. fifth condition), nor the difference between myc-Frat2 alone and myc-Frat2 + TMEM98-FLAG is statistically significant (i.e. the P value is >0.05). One explanation is the large intrinsic difference in the ability of Frat1 and Frat2 to induce TOPFLASH luciferase reporter activity, with Frat2 being a less potent activator of CTNNB1/TCF signaling (van Amerongen et al., Journal of Biological Chemistry, 2004 and van Amerongen et al., Oncogene, 2010). 

We chose to show only significant comparisons in the figure (as stated in the figure legend), which is why this P-value is not shown/depicted. The figures would get really cluttered if all non-significant comparisons are also included. 

We have modified the last sentence of the figure legend (page 15), which now states “Only statistically significant differences (one-way ANOVA, Tukey’s multiple comparisons test) are indicated with asterisks“. 

Please show the P value for the luciferase activity of myc-FRAT2 with and without TMEM98 RNAi #3 in Fig 4E. 

As above: 

When analyzed using a one-way ANOVA and Tukey’s multiple comparisons test (comparing every condition to every other condition), neither the difference between vector and myc-Frat2 alone (first condition vs. fifth condition), nor the difference between myc-Frat2 alone and myc-Frat2 + RNAi #3 is statistically significant (i.e. the P value is >0.05). We chose to show only significant comparisons in the figure (as stated in the figure legend), which is why this P-value is not shown/depicted. The figures would get really cluttered if all non-significant comparisons are also included. We have now modified the figure to include all significant comparisons (also Frat1 compared to vector, for instance) and we have changed the last sentence of the figure legend (page 16), which now states “Only statistically significant differences (one-way ANOVA, Tukey’s multiple comparisons test) are indicated with asterisks“. 

I am not convinced Fig 4F adds anything. Is it just a way of showing the data in Fig 4E in a different way?

The reviewer is correct, this is a different representation of the same data, as stated in the figure legend on page 16. 

Apologies but I could not find the figure legends for the supplementary figures in the revised submission.

Please note that the supplementary figure legends should have been included as a separate file (Supplementary Files.docx / Supplementary Files.pdf) with our last submission, since it was our impression that they should not be included with the main manuscript as per PLOS ONE manuscript preparation guidelines. 

This time they have again been uploaded as a separate file, again named Supplementary Files.docx / Supplementary Files.pdf

Reviewer #2: 

The reviewers have adequately addressed my questions and concerns, and explained why some experiments can not be done at this stage.

We thank Reviewer #2 for their time and effort.

---

## [Editor Report · Decision Letter 2]

19 Dec 2019

TMEM98 is a negative regulator of FRAT-mediated WNT/β-catenin signaling

PONE-D-19-18523R2

Dear Dr. van Amerongen,

We are pleased to inform you that your manuscript has been judged scientifically suitable for publication and will be formally accepted for publication once it complies with all outstanding technical requirements.

With kind regards,

Michael Koval

Academic Editor

PLOS ONE
---

## [Editor Report · Acceptance letter]

9 Jan 2020

PONE-D-19-18523R2 

TMEM98 is a negative regulator of FRAT mediated Wnt/β-catenin signalling 

Dear Dr. van Amerongen:

I am pleased to inform you that your manuscript has been deemed suitable for publication in PLOS ONE. Congratulations! Your manuscript is now with our production department. 

With kind regards,

on behalf of

Dr. Michael Koval 

Academic Editor

PLOS ONE